# Leveraging One-To-Many Relationships in Multimodal Adversarial Defense for Robust Image-Text Retrieval

## Abstract

Large pre-trained vision-language models (e.g., CLIP) are vulnerable to adversarial attacks in image-text retrieval (ITR). Existing works primarily focus on defense for image classification, overlooking two key aspects of ITR: multimodal manipulation by attackers, and the one-to-many relationship in ITR, where a single image can have multiple textual descriptions and vice versa (1:N and N:1). This is the first work that explores defense strategies for robust ITR. We demonstrate that our proposed multimodal adversarial training, which accounts for multimodal perturbations, significantly improves robustness against multimodal attacks; however, it suffers from overfitting to deterministic one-to-one (1:1) image-text pairs in the training data. To address this, we conduct a conprehensive study on leveraging one-to-many relationships to enhances robustness, investigating diverse augmentation techniques. Our findings reveal that diversity and alignment of image-text pairs are crucial for effective defense. Specifically, text augmentations outperform image augmentations, which tend to create either insufficient diversity or excessive distribution shifts. Additionally, we find that cross-modal augmentations (e.g., $image \rightarrow text$) can outperform intra-modal augmentations (e.g., $text \rightarrow text$) due to generating well-aligned image-text pairs. In summary, this work pioneers defense strategies for robust ITR, identifying critical aspects overlooked by prior research, and offers a promising direction for future studies.

## 1 Introduction

Image-text retrieval (ITR) is a fundamental Vision-Language (VL) task that involves cross-modal representation alignment between vision and language modalities. It consists of retrieving the most relevant text given an image query, and vice versa. One solution is using Vision-Language (VL) models pre-trained on large-scale paired image-text data, such as CLIP (Radford et al., 2021), which learns a joint embedding space for images and texts. However, recent studies revealed that all VL models for ITR are vulnerable to adversarial attacks (Zhang et al., 2022; Lu et al., 2023). Since adversarial attacks can deceive models with nearly negligible perturbations for humans, they pose significant risks of causing unintended consequences in real-world applications. For example, in e-commerce, retailers may add perturbations to the product images or descriptions to manipulate the retrieval results of an ITR system, unfairly promoting or demoting specific products. As the deployment of VL models in practical applications grows, understanding and mitigating their vulnerabilities against adversarial attacks has become crucial and urgent.

While several defense strategies for VL models have been proposed (Mao et al., 2022; Wang et al., 2024b; Schlarmann et al., 2024), they primarily focus on image attacks, e.g., for robust zero-shot image classification, leaving defense strategies tailored for ITR fully unexplored. This is a considerable oversight since ITR presents two challenges that make the problem much more complex compared to image classification: (1) *Multimodal manipulation*: Adversarial attacks on ITR can manipulate both image and text modalities, however, previous defense methods only consider image perturbations. Such an increased attack capability in ITR requires more complex defense strategies. (2) *One-to-many (1:N) cross-modal alignment*: Sentence-level text inputs in ITR exhibit a high degree of variation and ambiguity, as a single image can be described in numerous ways, and vice versa. The *one-to-many* nature of image-text alignment in ITR (e.g., a single image is described as

"a man with glasses is wearing a beer can crocheted hat" or "a man wears an orange hat and glasses") contrasts with simple and unambiguous text inputs in image classification tasks (e.g., all dog photos are paired with "a photo of a dog"), making harder to achieve a robust image-text alignment. Thus, the existing defense strategies for VL models aimed at image classification (Mao et al., 2022; Wang et al., 2024b) overlook these two critical aspects, casting doubt on their effectiveness for ITR.

To address this gap, we pioneer a study on defense strategies for VL models in ITR. Specifically, in this work, we study how to robustly fine-tune a large-scale vision-language model for downstream ITR tasks.

First, by incorporating multimodal perturbations during adversarial training, our proposed multimodal adversarial training (MAT) largely improves robustness against multimodal attacks. This highlights the need for defense strategies specifically tailored to multimodal threats in ITR, a requirement distinct vision-only defense strategies (Mao et al., 2022; Schlarmann et al., 2024).

However, we find that MAT suffers from overfitting to deterministic (1:1) image-text pairs in the training data. To mitigate this issue, we investigate how to consider the one-to-many (1:N) relationships in ITR to enhance adversarial robustness. Inspired by works in cross-modal ambiguity modeling in ITR (Kim et al., 2023; Song & Soleymani, 2019), we explore various text and image augmentation techniques to create diverse one-to-many (1:N) and many-to-one (N:1) image-text pairs. Our in-depth analysis reveal that diversity and alignment of image-text pairs are crucial for effective defense. For instance, text augmentations outperform image augmentations, which tend to create either insufficient diversity or excessive distribution shifts. Additionally, cross-modal augmentations (e.g., $image \rightarrow text$) can outperform intra-modal augmentations (e.g., $text \rightarrow text$) due to generating well-aligned image-text pairs. These findings are novel and unique for multimodal robustness settings, which has not been previously explored in unimodal adversarial training literature.

Our contributions are summarized as follows:

- **First defense strategy for ITR:** We demonstrate that existing defense methods for image classification are suboptimal for robust ITR, and pioneer research in this new direction.
- **Introduced multimodal adversarial training:** Our multimodal adversarial training largely improves robustness against multimodal attacks, highlighting the importance of considering multimodal perturbations for ITR defense.
- **Comprehensive analysis of leveraging one-to-many relationships for robust ITR:** We identify overfitting in multimodal adversarial training to deterministic one-to-one (1:1) image-text pairs. Thus, we provide an in-depth analysis of diverse augmentations, covering both image and text modalities, as well as intra- and cross-modal augmentations. We reveal that leveraging one-to-many (1:N) relationships improves robustness, with diversity and alignment of augmented image-text pairs being crucial for defense—insights not recognized in unimodal adversarial training literature.

## 2 RELATED WORK

**Adversarial attacks on vision-language models.** Adversarial attacks on VL models can be categorized into unimodal and multimodal attacks. Unimodal attacks, such as gradient-based image attacks (Madry et al., 2017) and BERT-Attack for text (Li et al., 2020), manipulate a single modality to mislead the models. On the other hand, recent studies have revealed that multimodal attacks, which perturb both image and text modalities, are significantly more effective (Zhang et al., 2022; Lu et al., 2023; Han et al., 2023; Wang et al., 2024a). However, developing defense strategies against multimodal attacks for ITR remains largely unexplored.

**Adversarial defense for vision-language models.** Existing defense strategies for VL models mainly focus on vision robustness, where only the image modality is perturbed by adversarial attacks. For example, Mao et al. (2022) and Wang et al. (2024b) have proposed robust fine-tuning methods of CLIP for zero-shot image classification, which leverage adversarial training scheme to improve the model's adversarial robustness. Schlarmann et al. (2024) proposed a method to robustly fine-tune a CLIP's vision encoder aimed at applications to diverse vision-language tasks, only focusing on image attacks. Unlike previous studies, we are the first to investigate adversarial defense

strategies for ITR tasks, where both image and text modalities can be manipulated by adversaries. Distinct from existing defense strategies, we propose multimodal adversarial training to improve robustness against multimodal attacks in ITR, and leverage the one-to-many (1:N) relationship in ITR to enhance adversarial robustness.

**Leveraging the one-to-many (1:N) nature of image-text.** To tackle robustness in VL models, we take inspiration from current works for ITR. These works aim to improve retrieval accuracy by modeling the ambiguity between image and text pairs; that is, although a sentence can have multiple visual interpretations, normally only one is paired as the ground truth. Similarly, an image can be described using multiple different captions, but only one is considered as its pair. Since such a 1:1 deterministic relationship is inconsistent with the 1:N nature of the data, ITR works propose representing image-text samples as probabilistic embeddings (Chun et al., 2021; Chun, 2024), considering neighboring samples in the triplet loss (Thomas & Kovashka, 2020), and generating multiple and diverse representations for each image-text sample (Song & Soleymani, 2019; Kim et al., 2023). Among these solutions, the latter naturally fits the data augmentation strategy of adversarial training. We hypothesize that leveraging data augmentation to increase diversity in a 1:N manner leads to robustness against adversarial attacks.

## 3 DEFENDING AGAINST VISION-LANGUAGE ADVERSARIAL ATTACKS

### 3.1 PRELIMINARIES

VL models Radford et al. (2021); Li et al. (2021); Yang et al. (2022) are fundamentally built on image-text contrastive learning, where the training objective is to maximize the similarity between matching image-text pairs while reducing the similarity between non-matching pairs in a shared embedding space. Among them, CLIP (Radford et al., 2021) is the representative VL model for ITR and is the foundation for many other VL models (Li et al., 2021; Yang et al., 2022). Thus, we focus on defense strategies for CLIP and conduct a comprehensive anlaysis on our proposed defense strategy. Below, we provide a brief overview of CLIP, followed by an explanations of existing adversarial attacks and defenses targeting CLIP.

**CLIP.** Contrastive Language-Image Pretraining (CLIP) consists of an image encoder $\Phi : \mathcal{I} \to \mathbb{R}^d$ and a text encoder $\Psi : \mathcal{T} \to \mathbb{R}^d$, where $\mathcal{I}$ and $\mathcal{T}$ are the input spaces for images and texts, respectively, and $d$ is the dimension of the joint embedding space. Given an image $I \in \mathcal{I}$ and a text $T \in \mathcal{T}$, CLIP is trained to embed them into the joint embedding space, and to maximize the similarity score $S_{\Phi,\Psi}(I, T) = sim(\Phi(I), \Psi(T))$ (cosine similarity of image-text embeddings) for correct image-text pairs, and minimize it for incorrect pairs. CLIP is based on image-text contrastive learning using the InfoNCE loss, where paired image-text samples form positive pairs, and unpaired image-text samples form negative pairs. For the batch of N paired image-text samples $\{(I_i, T_i)\}_{i=1}^N$, the InfoNCE loss (over images) is defined as:

$$\mathcal{L}_{CLIP_I} = \mathcal{L}_{CLIP_I}(I, T) = -\sum_{i=1}^N \log \frac{\exp(S_{\Phi,\Psi}(I_i, T_i)/\tau)}{\Sigma_{j=1}^N \exp(S_{\Phi,\Psi}(I_i, T_j)/\tau)}, \tag{1}$$

where $\tau$ is the learnable temperature parameter. The overall loss is the mean of the losses over images and texts, $\mathcal{L}_{CLIP} = (\mathcal{L}_{CLIP_I} + \mathcal{L}_{CLIP_T})/2$, where $\mathcal{L}_{CLIP_T}$ is the InfoNCE loss over texts.

**Multimodal adversarial attacks.** We aim to defend against adversarial attacks on CLIP for ITR, where both image and text modalities can be manipulated by adversarial attacks. The objective of (untargeted) adversarial attacks on CLIP is to minimize the image-text similarity $S_{\Phi,\Psi}(I, T)$ for the correct image-text pairs $(I, T)$ to mislead the models' predictions, as follows:

$$(I', T') = \underset{I', T'}{\arg\min}\, S_{\Phi,\Psi}(I', T'), \;\; s.t., \|I' - I\| \le \epsilon_I, \|T' - T\| \le \epsilon_T. \tag{2}$$

Image attacks add small perturbations to the original image $I$ to generate adversarial images $I'$, while maintaining perceptual similarity through an $L_p$-norm constraint $\|I' - I\|_p \le \epsilon_I$. A common method is the projected gradient descent (PGD) (Madry et al., 2017), which iteratively updates $I'$ by taking a small step in the direction of the gradient. Text attacks, such as BERT-Attack (Li et al., 2020), modify $N$ words in the text $T$ to maximize the divergence between $\Psi(T)$ and $\Psi(T')$. Multimodal attacks perturb both image-text modalities to generate adversarial examples $(I', T')$,

effectively combining image and text attacks to manipulate the image-text alignment. For example, Co-Attack (Zhang et al., 2022) perturbs both modalities in a step-wise manner, first perturbing the text, then the image given the perturbed text. SGA (Lu et al., 2023) enhances Co-Attack by considering the set-level interaction between the multiple images and texts.

**Adversarial defense of CLIP for zero-shot image classification.** The defacto standard defense strategy against adversarial attacks is adversarial training (AT) (Madry et al., 2017), which trains models using adversarial examples to improve robustness. To improve CLIP's adversarial robustness for zero-shot image classification tasks, TeCoA (Mao et al., 2022) adversarially fine-tunes the image encoder of CLIP by minimizing the contrastive loss between adversarial images and the text embeddings of the corresponding class labels, formulated as:

$$\mathcal{L}_{TeCoA} = \mathcal{L}_{CLIP_I}(I', T), \text{ where } I' = \arg\max_{I'} \mathcal{L}_{CLIP_I}(I', T), \tag{3}$$

where $I'$ is the adversarial image for the text $T$. However, TeCoA only defends against image adversarial attacks, and does not account for the one-to-many (1:N) relationship in ITR. Changing the previous paradigm, our work proposed a novel framework for robust fine-tuning of CLIP for ITR tasks, where both image and text modalities can be manipulated by attackers.

## 3.2 MULTIMODAL ADVERSARIAL TRAINING FOR MULTIMODAL ROBUSTNESS

We aim to defend against adversarial attacks on CLIP for ITR, which involves multimodal attacks that manipulate both image and text modalities. To this end, distinct from existing CLIP's adversarial fine-tuning methods (Mao et al., 2022; Schlarmann et al., 2024), which focus on vision robustness and fine-tune only CLIP's vision encoder, we start by fine-tuning the whole CLIP model to obtain robustness against multimodal perturbations.

To effectively defend against multimodal attacks, we propose a **m**ultimodal **a**dversarial **t**raining framework (MAT), which perturbs both image and text modalities during adversarial training. Here, we employ a step-by-step approach to generate adversarial examples $(I', T')$ by perturbing the image and text modalities sequentially. First, we generate adversarial texts $T'$ by maximizing the divergence between $\Psi(T)$ and $\Psi(T')$, formulated as:

$$T' = \arg\max_{T'} \|\Psi(T') - \Psi(T)\|, \text{ where } \|T' - T\| \leq \epsilon_T, \tag{4}$$

where the text perturbation is constrained by the number of words. Then, we generate adversarial images $I'$ by minimizing the cosine similarity score between $I'$ and the generated adversarial text $T'$ as follows:

$$I' = \arg\max_{I'} -\frac{\Phi(I') \cdot \Psi(T')}{\|\Phi(I')\|\|\Psi(T')\|}, \text{ where } \|I' - I\| \leq \epsilon_I, \tag{5}$$

where the image perturbation is constrained by $L_p$-norm with $\epsilon_I$. Finally, we update the model parameters by minimizing the contrastive loss between the adversarial image $I'$ and the adversarial text $T'$, formulated as:

$$\mathcal{L}_{\mathcal{MAT}} = \mathcal{L}_{CLIP}(I', T'). \tag{6}$$

To implement the text perturbation during fine-tuning, we use the BERT-Attack, which iteratively replaces important words in the text to maximize the divergence between $\Psi(T)$ and $\Psi(T')$. In the case of images, we utilize the widely adopted PGD to generate adversarial image perturbations.

In Section 4.3, we show that multimodal adversarial fine-tuning significantly improves robustness against multimodal attacks in ITR, compared to existing defense strategies that only consider image perturbations. Nevertheless, we find that this framework easily overfits to deterministic pairs in the training data, which we discuss in the following section.

## 3.3 LEVERAGING ONE-TO-MANY IMAGE-TEXT PAIRS FOR ROBUSTNESS GENERALIZATION

Existing works on cross-modal ambiguity in ITR (Kim et al., 2023; Song & Soleymani, 2019) demonstrated the importance of modeling the one-to-many (1:N) relationship that exists between image and text descriptions. In our setting, a simple adversarial fine-tuning on a downstream ITR

dataset would also easily overfit to deterministic (1:1) image-text pairs in the training data. Therefore, the inherent ambiguity of image-text pairs in ITR needs to be considered to achieve a more optimal adversarial robustness in ITR.

To this end, we propose Multimodal Augmented Adversarial Training (MA$^2$T), leveraging data augmentation strategies to create diverse one-to-many (1:N) and many-to-one (N:1) image-text pairs to prevent overfitting in MAT. Our idea is streightforward: a single image can be described in numerous ways, and vice versa. Text augmentation can generate diverse text samples for a given image, creating 1:N image-text pairs. Similarly, image augmentation can generate diverse images for a given text, creating N:1 image-text pairs. In this way, we can prevent overfitting to deterministic (1:1) pairs and naturally model the ambiguity of image-text pairs in ITR during mutlimodal adversarial fine-tuning, improving adversarial robustness.

Thus, given an image-text pair $(I, T)$, we generate augmented images $I_{aug} \leftarrow aug_I(I, T)$ or augmented text $T_{aug} \leftarrow aug_T(I, T)$, where $aug_I$ and $aug_T$ are image and text augmentation functions, respectively. Then, the new augmented pairs $(I_{aug}, T)$ or $(I, T_{aug})$ are added to the training data for the image or text augmentation scenarios. Note that we do not use the pairs $(I_{aug}, T_{aug})$ in the training process, as $I_{aug}$ and $T_{aug}$ are not necessarily aligned with each other since they are generated independently. Using these pairs empirically degrades the model's performance.

The proposed defense strategy is summarized in Algorithm 1.

---

**Algorithm 1** Multimodal Augmented Adversarial Training (MA$^2$T)

---

**Require:** Image-text pairs $(I, T) \sim D$, Model $\theta$, Learning rate $\alpha$, Perturbation constraints $\epsilon_I, \epsilon_T$
1: **(Data Preparation:)**
2: **for** each $(I, T) \in D$ **do**
3:     **(N:1 case)** Image augmentation: $I^{aug} \leftarrow aug_I(I, T), D \leftarrow D \cup (I^{aug}, T)$
4:     **(1:N case)** Text augmentation: $T^{aug} \leftarrow aug_T(I, T), D \leftarrow D \cup (I, T^{aug})$
5: **end for**
6: **(Training:)**
7: **for** each batch **do**
8:     $T' \leftarrow \arg\max_{T'} \|\Psi(T') - \Psi(T)\|$, where $\|T' - T\| \leq \epsilon_T$
9:     $I' \leftarrow \arg\max_{I'} - \frac{\Phi(I') \cdot \Psi(T')}{\|\Phi(I')\|\|\Psi(T')\|}$, where $\|I' - I\| \leq \epsilon_I$
10:     $\theta \leftarrow \theta - \alpha \cdot \nabla_\theta \mathcal{L}_{CLIP}(I', T')$
11: **end for**

---

However, deciding which types of augmentations are more effective is not trivial, since these can model either intra- or cross-modal relationships, and be more or less computationally complex. The next section thoroughly explores which augmentations are more effective and why.

## 4 EXPERIMENTS

### 4.1 EXPERIMENTAL SETTINGS

**Datasets.** We employ the Flickr30k (Plummer et al., 2015) and COCO (Chen et al., 2015) datasets, which are widely used for ITR. We use the default train/test split of 29,000/1,000 images for Flickr30k, and 82,783/40,775 images for COCO. While Flickr30k and COCO contain five captions per image, our baseline training approach uses 1:1 image-text pairs, as this is the practical setting for fine-tuning CLIP. Thus, when creating 1:1 pairs, we take the first annotated caption of each image.

**Evaluation.** We evaluate our proposed defense strategies against adversarial attacks in the ITR task using the Recall@$k$ (R@$k$) metric. This includes both image-to-text retrieval (TR) and text-to-image retrieval (IR), where the objective is to retrieve the most relevant text for a given image query and vice versa. We employ multimodal adversarial attacks, Co-Attack, and SGA, which are more effective at deceiving VL models than unimodal attacks. The perturbation constraints are set to $\epsilon_I = 2/255$ with $L_\infty$-norm for image attacks, and one word for text attacks. An evaluation for unimodal attacks, including PGD and BERT-Attack, is provided in Appendix A.2.

Table 1: Comparison of defense methods for clean (i.e., no attack), Co-Attack, and SGA scenarios.

|  | Method | Clean | | | | Co-Attack | | | | SGA | | | |
|---|---|---|---|---|---|---|---|---|---|---|---|---|---|
|  |  | TR | | IR | | TR | | IR | | TR | | IR | |
|  |  | R@1 | R@5 | R@1 | R@5 | R@1 | R@5 | R@1 | R@5 | R@1 | R@5 | R@1 | R@5 |
| Flickr | Fine-tune | 92.1 | 99.0 | 77.2 | 94.4 | 11.0 | 23.3 | 6.7 | 16.5 | 0.6 | 2.0 | 0.6 | 2.6 |
|  | TeCoA | 81.3 | 94.7 | 67.6 | 89.0 | 53.3 | 79.9 | 35.9 | 61.2 | 31.6 | 60.4 | 22.1 | 46.0 |
|  | (ours) MAT$_{Img}$ | 84.8 | 96.3 | 68.7 | 89.4 | 52.8 | 77.7 | 31.6 | 56.8 | 27.3 | 52.9 | 17.9 | 40.0 |
|  | (ours) MAT | 81.8 | 95.4 | 67.0 | 88.0 | 55.0 | 80.0 | 35.8 | 62.3 | 33.9 | 60.7 | 23.0 | 47.7 |
| COCO | Fine-tune | 66.6 | 88.0 | 50.1 | 76.5 | 2.9 | 7.5 | 1.8 | 5.1 | 0.1 | 0.5 | 0.1 | 0.6 |
|  | TeCoA | 51.5 | 77.5 | 35.1 | 61.9 | 20.6 | 43.8 | 12.6 | 29.1 | 10.1 | 25.5 | 7.4 | 19.3 |
|  | (ours) MAT$_{Img}$ | 59.2 | 83.0 | 42.0 | 69.9 | 21.4 | 45.3 | 12.3 | 29.0 | 10.0 | 24.6 | 6.2 | 17.5 |
|  | (ours) MAT | 55.6 | 80.5 | 40.2 | 68.1 | 29.2 | 55.6 | 18.4 | 40.9 | 15.7 | 35.4 | 11.0 | 27.6 |

**Comparison methods.** Our methods are compared with the baseline TeCoA, which fine-tunes only the vision encoder of CLIP for zero-shot image classification. We also evaluate two variants of our approach: (1) MAT$_{Img}$, which perturbs only the image modality using PGD, and (2) MAT, which perturbs both modalities using PGD and BERT-Attack.

**Training details.** We use the pretrained CLIP-ViT-B/16 (Radford et al., 2021) as the base model to adversarially fine-tune. We fix the total number of training steps to 5,000, and the batch size to 128 for all experiments. We use the SGD optimizer with cosine learning rate scheduling, where the initial learning rate is set to 0.0001, and the weight decay is set to 0.0001.

## 4.2 AUGMENTATION STRATEGIES

**Augmentation types: Intra-modal and Cross-modal.** We consider two types of augmentation techniques: *intra-modal* and *cross-modal*. Intra-modal augmentation enhances data points without considering image-text interactions (i.e., text → text, image → image). For example, basic image augmentation, such as random cropping, corresponds to intra-modal augmentation, as it does not require any knowledge of the paired text data. In contrast, cross-modal augmentation enhances data points by leveraging the other modality (i.e., image → text, text → image). An example is generating plausible images from a given caption via a text-to-image generative model.

**Text augmentations.** We consider two text augmentation techniques for intra-modal augmentation: Easy Data Augmentation (EDA) (Wei & Zou, 2019) and Language Rewrite (LangRW) (Fan et al., 2024). EDA is a simple text augmentation technique that applies four types of operations: synonym replacement, random insertion, random deletion, and random swap. LangRW is a more advanced text augmentation technique that leverages a large language model to rewrite the original texts. For cross-modal augmentation, we consider an image-to-text generative model, OFA (Wang et al., 2022), which generates plausible captions from a given image. Additionally, we consider human annotations, denoted as *Human*, as cross-modal augmentation: Flickr30k and COCO contain five captions per image, so we use the remaining four captions as augmented data points.

**Image augmentations.** For intra-modal augmentation, we consider two image augmentation techniques: For intra-modal augmentation, RandAugment (RandAug; RA) (Cubuk et al., 2020). RandAug applies a series of image augmentations, such as random cropping, color distortion, and rotation, to the original image. For cross-modal augmentation, we consider a text-to-image generative model, Stable Diffusion (SD) (Rombach et al., 2022), which generates plausible images from a given caption.

**Augmentation settings.** For a fair comparison, we fix the number of augmented data points to be five times the number of original data points (generating four augmented data points for each original data point). Please refer to the Appendix A.1 for the detailed settings of each augmentation technique.

## 4.3 EFFECTIVENESS OF MULTIMODAL AUGMENTED ADVERSARIAL TRAINING (MA$^2$T)

**Effectiveness of multimodal adversarial training (MAT).** Table 1 compares defense strategies against clean (i.e., no adversarial attack), Co-Attack, and SGA scenarios on Flickr30k and COCO.

Table 3: Effectiveness of augmentation techniques for $MA^2T$ on Flickr30k. We compare our methods with and without augmentation, highlighting the best performance boost for each MAT method in bold text. T2T and I2I (i.e., text-to-text and image-to-image) denote intra-modal augmentations, while T2I and I2T (i.e., text-to-image and image-to-text) denote cross-modal augmentations. The results demonstrate the importance of leveraging 1:N and N:1 image-text pairs for further improving robustness.

| | Img aug. | Text. aug. | Co-Attack | | | | SGA | | | |
| | | | TR | | IR | | TR | | IR | |
| | | | R@1 | R@5 | R@1 | R@5 | R@1 | R@5 | R@1 | R@5 |
|---|---|---|---|---|---|---|---|---|---|---|
| Fine-tune | | | 11.0 | 23.3 | 6.7 | 16.5 | 0.6 | 2.0 | 0.6 | 2.6 |
| TeCoA | | | 55.2 | 80.7 | 33.1 | 58.8 | 33.6 | 61.1 | 21.7 | 44.9 |
| (ours) $MAT_{Img}$ | | | 52.8 | 77.7 | 31.6 | 56.8 | 27.3 | 52.9 | 17.9 | 40.0 |
| | | T2T(EDA) | 56.2 | **81.2** ↑3.5 | 34.8 | 61.0 | 32.7 | 58.1 | 21.1 | 45.6 |
| + 1:N Aug | | T2T(LangRW) | 53.9 | 78.5 | 33.3 | 59.0 | 31.5 | 56.1 | 21.0 | 43.7 |
| | | I2T(OFA) | **56.6** ↑3.8 | 81.0 | 34.6 | 59.5 | 33.2 | 59.4 | 21.6 | 45.5 |
| | | I2T(Human) | 56.6 | 80.4 | **35.7** ↑4.0 | **61.3** ↑4.4 | **34.7** ↑7.4 | **60.0** ↑7.1 | **22.4** ↑4.5 | **46.9** ↑6.9 |
| + N:1 Aug | I2I(RA) | | 52.7 | 77.2 | 31.4 | 55.7 | 27.1 | 53.3 | 18.1 | 38.9 |
| | T2I(SD) | | 50.9 | 76.5 | 31.5 | 57.4 | 29.7 | 54.8 | 18.7 | 41.3 |
| (ours) MAT | | | 55.0 | 80.0 | 35.8 | 62.3 | 33.9 | 60.7 | 23.0 | 47.7 |
| | | T2T(EDA) | 58.1 | 81.7 | 39.3 | 66.0 | 37.4 | 63.8 | 26.2 | 52.5 |
| + 1:N Aug | | T2T(LangRW) | 58.7 | 81.4 | 40.1 | 66.2 | 38.2 | 62.5 | 26.8 | 53.4 |
| | | I2T(OFA) | 58.4 | 83.2 | 39.9 | 66.2 | 38.2 | 67.3 | 27.5 | 53.9 |
| | | I2T(Human) | **63.7** ↑8.7 | **83.9** ↑3.9 | **42.4** ↑6.6 | **68.7** ↑6.4 | **44.4** ↑10.5 | **68.8** ↑8.1 | **30.8** ↑7.8 | **56.7** ↑9.0 |
| + N:1 Aug | I2I(RA) | | 56.0 | 80.3 | 36.4 | 61.7 | 33.7 | 62.5 | 23.0 | 47.5 |
| | T2I(SD) | | 55.7 | 79.7 | 36.7 | 63.3 | 34.9 | 60.7 | 23.7 | 48.8 |

Fine-tuning is the case where no defense strategy is applied. By incorporating multimodal adversarial perturbations, MAT outperforms both $MAT_{Img}$ and TeCoA, which only considers image perturbations, demonstrating the effectiveness of multimodal adversarial training for ITR. However, $MAT_{Img}$ achieves slightly worse performance than TeCoA in the attacked scenarios, highlighting the limitations of traditional unimodal paradigms when applied to multimodal settings. The next section further explores the challenge of adversarial fine-tuning the entire CLIP model for ITR tasks due to overfitting.

**Overfitting issue in Multimodal adversarial training (MAT).** In Table 2, we observe that the adversarial fine-tuning for ITR easily overfits to the train data, leading to poor robustness generalization to unseen data. For example, MAT on Flickr30k shows a performance gap of 18.0% and 35.6% in TR@5 and IR@5, respectively, between the train and test sets. On the other hand, $MA^2T$, using augmented texts by human annotations, significantly mitigates the overfitting issue, reducing the performance gap to 8.9% and 11.6% in TR@5 and IR@5, respectively.

Table 2: Overfit issue of MAT on Flickr30k. We report the performances on the train and test sets against Co-Attack.

| | MAT | | $MA^2T_{I2T(Human)}$ | |
| | TR@5 | IR@5 | TR@5 | IR@5 |
|---|---|---|---|---|
| Train | 98.0 | 97.9 | 92.8 | 80.3 |
| Test | 80.0 | 62.3 | 83.9 | 68.7 |
| *Diff* | *-18.0* | *-35.6* | *-8.9* | *-11.6* |

**Effectiveness of $MA^2T$: Augmenting image-text pairs improve robustness.** Table 3, summarizes the effectiveness of one-to-many and many-to-one augmentations in improving adversarial robustness on Flickr30k. We find that effectively leveraging augmentations can further improve adversarial robustness of MAT. In the following sections, we analyze the factors that contribute to the effectiveness of the augmentation techniques in $MA^2T$.

### 4.4 LEVERAGING ONE-TO-MANY RELATIONSHIPS: A COMPREHENSIVE ANALYSIS

To understand the effectiveness of leveraging the one-to-many relationship in robust ITR, we analyze the properties of the augmented data points generated by the augmentation techniques. To this end, we analyze the alignment quality and diversity of the augmented image-text pairs. Figure 1 shows the distribution of alignment scores of augmented image-text pairs, where the alignment score is calculated as the cosine similarity of the embeddings of the image and text pairs, measured in a pretrained CLIP's joint embedding space. Figure 2 presents the distribution of L2 distance of embeddings before and after augmentation, with higher L2 distances indicating greater augmentation diversity.

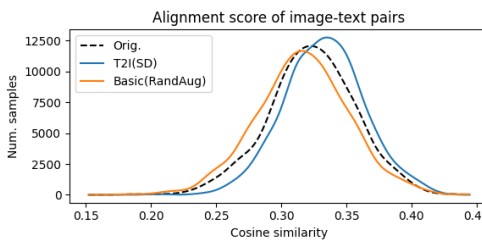 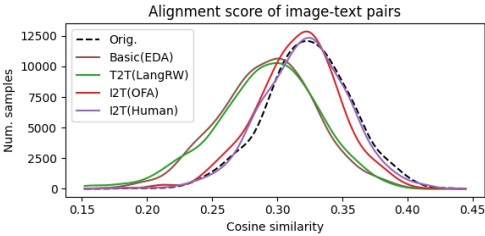

(a) Alignment analysis for image augmentation.     (b) Alignment analysis for text augmentation.

Figure 1: Alignment measures the similarity between an image (or text) sample and its augmented text (or image) pair (cross-modal) in the Flickr30k dataset. We denote the original 1:1 image-text pairs as "Orig." We plot the distribution of the alignment scores of the augmented image-text pairs (cosine similarity between image and text embeddings), measured in the pretrained CLIP's embedding space as: $sim(\Phi(aug_I(I)), \Psi(T))$ or $sim(\Phi(I), \Psi(aug_T(T)))$, where $\Phi$ and $\Psi$ are image (I) and text (T) encoders of CLIP, respectively. A higher alignment score indicates that augmentations are better aligned with their corresponding cross-modal pair, suggesting better augmentation quality.

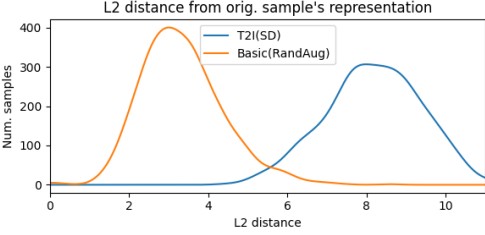 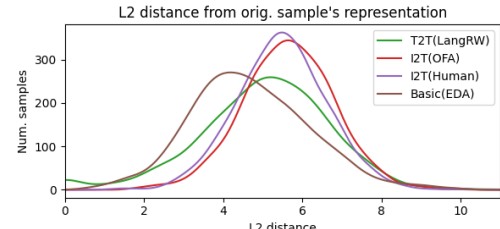

(a) Diversity analysis for image augmentation.     (b) Diversity analysis for text augmentation.

Figure 2: Diversity measures the distance between an image (or text) sample and its augmented image (or text) (uni-modal) in the Flickr30k dataset. We plot the distribution of the L2 distance between the embeddings before and after augmentation, measured in the pretrained CLIP embedding space as: $L2(\Phi(I), \Phi(aug_I(I)))$ or $L2(\Psi(T), \Psi(aug_T(T)))$, where $\Phi$ and $\Psi$ are image (I) and text (T) encoders of CLIP, respectively. A higher L2 distance indicates that the generated data points are more distant from the original ones, suggesting greater augmentation diversity.

**Cross-modal augmentation surpasses intra-modal augmentation due to better alignment of augmented image-text pairs.** In Table 3, we observe that cross-modal augmentations, such as OFA and Human, can outperform intra-modal augmentations, such as EDA and LangRW, in improving adversarial robustness. This is explained by the alignment analysis in Figure 1, where cross-modal augmentations generate more aligned image-text pairs compared to intra-modal augmentations: while both EDA and LangRW improved adversarial robustness, the alignment scores of the augmented image-text pairs are lower compared to cross-modal augmentations, such as OFA and Human, which achieved better alignment scores and adversarial robustness. In the case of images, although cross-modal augmentations are qualitatively superior, alignment does not vary drastically.

**Diverse and well-aligned image-text pairs lead to better robustness.** Additionally, we observed that the balance of alignment and diversity is crucial for the effectiveness of the defense strategy in improving adversarial robustness. In Figure 2, we observe that intra-modal augmentations, like RandAug for images and EDA and LangRW for text, generate less diverse image-text pairs because they only slightly modify the original data points, mostly preserving their semantics. In contrast, cross-modal augmentations, like SD for image and OFA and Human for text, generate more diverse image-text pairs that are more distant from the original data points. This is due to the inherent uncertainty in cross-modal generation caused by the one-to-many relationship in ITR, where a single image can be described in several ways and vice versa, which naturally increases the diversity of the augmented data. This property of cross-modal augmentations can lead to better adversarial

Table 4: Effectiveness of augmentation techniques for our methods on COCO. Well-aligned and diverse augmentations (e.g., I2T(Human)) consistently provide performance gains, demonstrating the consistent effectiveness of augmentations across different datasets.

| | Img aug. | Text. aug. | Co-Attack | | | | SGA | | | |
|---|---|---|---|---|---|---|---|---|---|---|
| | | | TR | | IR | | TR | | IR | |
| | | | R@1 | R@5 | R@1 | R@5 | R@1 | R@5 | R@1 | R@5 |
| Fine-tune | | | 2.9 | 7.5 | 1.8 | 5.1 | 0.1 | 0.5 | 0.1 | 0.6 |
| TeCoA | | | 20.6 | 43.8 | 12.6 | 29.1 | 10.1 | 25.5 | 7.4 | 19.3 |
| (ours) $\text{MAT}_{\text{Img}}$ | | | 21.4 | 45.3 | 12.3 | 29.0 | 10.0 | 24.6 | 6.2 | 17.5 |
| + 1:N Aug | | I2T(OFA) | 22.7 | 46.2 | 12.8 | 29.3 | 11.2 | 27.0 | 6.9 | 18.4 |
| | | I2T(Human) | 23.5 ↑2.1 | 47.1 ↑1.8 | 13.3 ↑1.1 | 30.6 ↑1.6 | 11.5 ↑1.4 | 27.8 ↑3.3 | 7.0 ↑0.8 | 19.3 ↑1.8 |
| + N:1 Aug | T2I(SD) | | 18.4 | 40.0 | 10.9 | 26.1 | 8.2 | 20.8 | 5.1 | 15.2 |
| (ours) MAT | | | 29.2 | 55.6 | 18.4 | 40.9 | 15.7 | 35.4 | 11.0 | 27.6 |
| + 1:N Aug | | I2T(OFA) | 28.2 | 54.0 | 18.0 | 39.5 | 16.0 | 35.2 | 11.3 | 27.5 |
| | | I2T(Human) | 30.9 ↑1.7 | 57.4 ↑1.8 | 19.7 ↑1.2 | 42.2 ↑1.3 | 17.9 ↑2.2 | 37.9 ↑2.4 | 12.0 ↑1.1 | 29.9 ↑2.3 |
| + N:1 Aug | T2I(SD) | | 25.5 | 49.6 | 16.8 | 36.9 | 13.1 | 30.1 | 9.5 | 24.5 |

robustness, as shown in Table 3 that OFA and Human outperform RandAug and EDA/LangRW, respectively.

**Efficacy of text and image augmentations.** In Table 3, we found an efficacy gap between text and image augmentations, providing the latter higher boosts. While SD generates diverse image-text pairs with high alignment scores, it does not improve adversarial robustness as much as OFA and Human. This is due to a large distribution gap between the generated and the original images, which leads to a distribution shift that degrades the model's performance. Additionally, in Figure 3a, we plot the robustness performance of $\text{MAT}_{\text{Img}}$ with SD on Flickr30k against SGA, where the number of additional images used for adversarial training is varied. Top indicates they are sorted decreasingly by alignment score. We find that the adversarial robustness of $\text{MAT}_{\text{Img}}$ with SD improves when up to two additional images are used, but beyond that, the performance starts to degrade. This suggests that the effective defense strategy should employ augmentations that generate "moderately" diverse image-text pairs that do not introduce a significant distribution shift. In comparison, in Table 3, we observe that text augmentations, such as Human and OFA, can significantly improve adversarial robustness. For example, Figure 3b illustrates that increasing the number of Human augmentations consistently boosts robustness. This is because generating image augmentations that do not lack diversity but also do not deviate significantly from the original data distribution is more challenging due to the high dimensionality of the image space. On the other hand, text modality is more amenable to augmentation, as the text space is lower-dimensional and more structured, making it easier to generate appropriate diversity in the augmented data points.

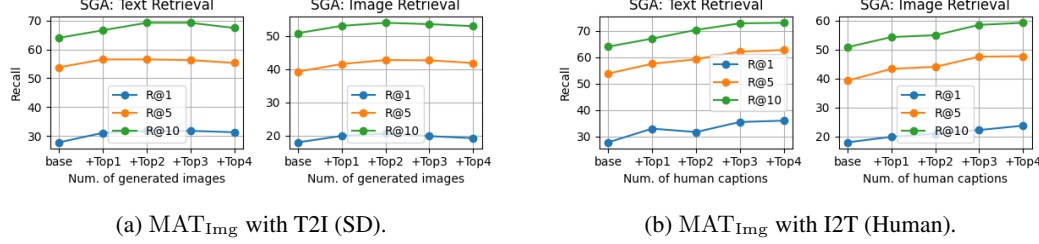

(a) $\text{MAT}_{\text{Img}}$ with T2I (SD).  (b) $\text{MAT}_{\text{Img}}$ with I2T (Human).

Figure 3: Effectiveness of cross-modal augmentations, SD and Human, in improving adversarial robustness in VL models for ITR.

**Evaluation on COCO dataset.** We also analyze augmentations on the COCO dataset in Table 4. We find that only the well-aligned and diverse augmentations, I2T(Human), consistently provide performance gains on COCO. This is because COCO has a larger number of training samples compared to Flickr30k, and the improvements by augmentations are less significant. However, the gains from I2T(Human) suggest that the effectiveness of well-aligned and diverse augmentations remains consistent across datasets, with performance significantly surpassing that of TeCoA.

## 5    LIMITATIONS

We focused on CLIP as the base model for our defense strategy in order to deeply analyze the effectiveness of our framework, leaving the exploration of other vision-language models for future work. Additionally, to improve our framework, a method that generates image augmentations that do not create a distribution shift from the original data should be proposed.

## 6    CONCLUSIONS

This is the first work to study adversarial defense strategies for vision-language (VL) models in the context of image-text retrieval (ITR). Existing defense methods for CLIP are not effective against multimodal attacks, as they are designed to defend against image-only attacks and do not consider the one-to-many (1:N) nature of images and texts. For this reason, we proposed a novel defense framework, Multimodal Augmented Adversarial Training ($\text{MA}^2\text{T}$), which leverages one-to-many (1:N) image-text pairs via augmentations to enhance robustness for ITR. Our comprehensive analysis reveals that our framework can significantly improve adversarial robustness against multimodal attacks, and that well-aligned and diverse augmentations are crucial for effective defense, which was previously unexplored in the literature on unimodal adversarial defense. This work identifies novel challenges overlooked in previous works and provides a novel perspective on adversarial defense strategies for VL models in ITR, fostering future research for reliable and secure AI systems.

**Ethics Statement.** In this work, we focus on improving the robustness of vision-language models for Image-Text Retrieval (ITR) tasks against adversarial attacks. We use publicly available datasets (Flickr30k and COCO) with no human subjects or personal data. We acknowledge potential biases in pre-trained models like CLIP and emphasize that our work is aimed at enhancing security and reliability, not harmful applications. While adversarial methods can be misused, our research is focused on defense strategies, promoting secure AI systems. Our work complies with legal and ethical standards, with no conflicts of interest, and we ensure research transparency by making our methods and findings publicly available for reproducibility.

**Reproducibility.** We have made extensive efforts to ensure the reproducibility of our results. Detailed descriptions of our experimental setups, including model architectures, hyperparameters, and training protocols, are provided in the main text and appendix. We also offer a comprehensive explanation of data processing steps for the Flickr30k and COCO datasets. All algorithms, including adversarial fine-tuning methods, are described in detail, and additional implementation details are included in the supplementary materials. For further reproducibility, we will provide anonymous access to the source code, which will be included in the supplementary materials.

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

# A    APPENDIX / SUPPLEMENTAL MATERIAL

## A.1    IMPLEMENTATION DETAILS OF AUGMENTATION TECHNIQUES

### A.1.1    TEXT AUGMENTATIONS

**EDA (Easy Data Augmentation).** EDA randomly selects words in the text and performs the following operations: synonym replacement, random insertion, random swap, or random deletion. We use the official implementation [1]. The hyperparameter $\alpha$ controls the strength of the augmentation, where $\alpha$ determines the probability of each word being augmented. We use $\alpha = 0.3$ for all experiments.

**LangRW (Language rewrite).** Language rewrite (LangRW) (Fan et al., 2024) is a method that rewrites the text data to improve the robustness of the model, using a generative natural language processing model, such as Llama (Touvron et al., 2023). We used Llama-2-7B [2]. In our work, we used slightly modified prompts from the original work to simultaneously generate four captions per image. Given an original caption $T$, the prompt for generating additional captions are as follows:

```
Rewrite image captions in 4 different ways.

{coco caption 1 for image i}
=> {coco caption 2 for image i}
=> {coco caption 3 for image i}
=> {coco caption 4 for image i}
=> {coco caption 5 for image i}

{coco caption 1 for image j}
=> {coco caption 2 for image j}
=> {coco caption 3 for image j}
=> {coco caption 4 for image j}
=> {coco caption 5 for image j}

{coco caption 1 for image k}
=> {coco caption 2 for image k}
=> {coco caption 3 for image k}
=> {coco caption 4 for image k}
=> {coco caption 5 for image k}

{original caption to be rewritten}
=>
```

where the coco captions are randomly sampled from the original captions from the COCO dataset (Chen et al., 2015).

**OFA.** OFA (Wang et al., 2022) is an image captioning model. We use the official implementation [3]. We used the default prompt of " what does the image describe?" to generate additional captions for each image.

**Human.** Human augmentation is a method that generates additional captions by human annotators. Since we use 1:1 image-text pairs for training as default, we used the rest of the original captions included in Flickr30k and COCO datasets as additional captions for each image.

### A.1.2    IMAGE AUGMENTATIONS

**RandAug (Random Augmentation).** RandAug (Cubuk et al., 2020) is an image augmentation method that applies a series of random transformations to the image. We used the codes from

---

[1]https://github.com/jasonwei20/eda_nlp

[2]https://huggingface.co/meta-llama/Llama-2-7b

[3]https://github.com/OFA-Sys/OFA

ALBEF (Li et al., 2021) [4]. We set the number of operations to 2 and the magnitude to 5 for all experiments.

**Stable Diffusion (SD).** Stable Diffusion (SD) (Rombach et al., 2022) is a text-to-image generative model. We used SD-v2.1 [5].

## A.2 Evaluation of MA$^2$T on other attack types: Clean, PGD, and BERT

In this section, we present the evaluation results of the proposed MA$^2$T framework against different attack types, including clean, PGD, and BERT attacks.

Table 5: Effectiveness of augmentation techniques for MA$^2$T on Flickr30k for different attack types, including clean, PGD, and BERT attacks.

| | Img aug. | Text. aug. | Clean | | | | PGD | | | | BERT-Attack | | | |
|---|---|---|---|---|---|---|---|---|---|---|---|---|---|---|
| | | | TR | | IR | | TR | | IR | | | | | |
| | | | R@1 | R@5 | R@1 | R@5 | R@1 | R@5 | R@1 | R@5 | R@1 | R@5 | R@1 | R@5 |
| Finetune | | | 92.1 | 99.0 | 77.2 | 94.4 | 0.7 | 2.0 | 0.7 | 2.1 | 75.4 | 93.4 | 53.1 | 78.3 |
| TeCoA | | | 81.6 | 95.5 | 68.2 | 89.3 | 52.7 | 78.7 | 43.4 | 70.5 | 68.2 | 90.1 | 46.9 | 72.0 |
| (ours) MAT$_{Img}$ | | | 84.8 | 96.3 | 68.7 | 89.4 | 51.3 | 76.2 | 40.1 | 69.4 | 66.2 | 90.2 | 43.4 | 69.8 |
| | | T2T(EDA) | 85.6 | 95.8 | 70.9 | 90.4 | 54.4 | 78.0 | 43.2 | 71.7 | 71.4 | 91.5 | 47.8 | 73.3 |
| + 1:N Aug | | T2T(LangRW) | 83.8 | 94.9 | 69.3 | 89.5 | 54.2 | 75.6 | 43.3 | 70.6 | 65.8 | 88.3 | 44.2 | 70.5 |
| | | I2T(OFA) | 85.9 | 96.3 | 69.8 | 90.1 | 59.9 | 81.1 | 45.4 | 73.2 | 68.1 | 89.4 | 44.5 | 70.7 |
| | | I2T(Human) | 86.7 | 97.3 | 73.1 | 92.6 | 58.1 | 80.7 | 48.2 | 74.8 | 69.6 | 90.9 | 47.0 | 73.7 |
| + N:1 Aug | I2I(RA) | | 85.0 | 96.8 | 67.4 | 88.6 | 54.0 | 76.3 | 40.6 | 69.1 | 65.8 | 88.8 | 41.6 | 67.8 |
| | T2I(SD) | | 85.7 | 96.6 | 70.1 | 90.9 | 50.8 | 74.8 | 39.0 | 67.3 | 67.8 | 89.8 | 45.6 | 71.5 |
| (ours) MAT | | | 81.8 | 95.4 | 67.0 | 88.0 | 53.8 | 77.2 | 39.4 | 68.3 | 70.1 | 91.1 | 50.1 | 74.5 |
| | | T2T(EDA) | 85.6 | 96.4 | 69.3 | 89.3 | 56.1 | 78.8 | 43.3 | 71.6 | 73.4 | 93.1 | 52.1 | 78.4 |
| + 1:N Aug | | T2T(LangRW) | 81.9 | 94.8 | 67.5 | 88.5 | 52.5 | 76.2 | 42.6 | 70.4 | 73.5 | 91.9 | 51.3 | 77.4 |
| | | I2T(OFA) | 84.7 | 95.3 | 68.1 | 89.4 | 58.9 | 80.9 | 44.8 | 72.7 | 72.4 | 92.5 | 51.6 | 76.9 |
| | | I2T(Human) | 86.3 | 96.5 | 71.4 | 91.0 | 58.3 | 80.5 | 46.7 | 74.9 | 77.2 | 93.0 | 55.8 | 80.9 |
| + N:1 Aug | I2I(RA) | | 83.3 | 95.2 | 66.0 | 87.5 | 54.5 | 78.1 | 39.3 | 68.0 | 70.8 | 91.4 | 50.0 | 74.8 |
| | T2I(SD) | | 83.2 | 95.8 | 67.9 | 89.6 | 50.9 | 74.9 | 39.9 | 67.6 | 70.6 | 92.1 | 51.0 | 76.3 |

Table 6: Effectiveness of augmentation techniques for MA$^2$T on COCO for different attack types, including clean, PGD, and BERT attacks.

| | Img aug. | Text. aug. | Clean | | | | PGD | | | | BERT-Attack | | | |
|---|---|---|---|---|---|---|---|---|---|---|---|---|---|---|
| | | | TR | | IR | | TR | | IR | | | | | |
| | | | R@1 | R@5 | R@1 | R@5 | R@1 | R@5 | R@1 | R@5 | R@1 | R@5 | R@1 | R@5 |
| Finetune | | | 66.6 | 88.0 | 50.1 | 76.5 | 0.2 | 0.9 | 0.2 | 0.7 | 36.9 | 65.2 | 23.7 | 46.7 |
| TeCoA | | | 51.5 | 77.5 | 35.1 | 61.9 | 29.2 | 53.6 | 20.0 | 42.1 | 29.3 | 55.5 | 17.6 | 37.5 |
| (ours) MAT$_{Img}$ | | | 59.2 | 83.0 | 42.0 | 69.9 | 31.4 | 55.2 | 21.7 | 46.4 | 31.0 | 58.5 | 18.5 | 39.4 |
| + 1:N Aug | | I2T(OFA) | 58.7 | 82.3 | 41.4 | 69.1 | 32.8 | 57.0 | 22.4 | 47.0 | 31.6 | 58.6 | 18.6 | 38.8 |
| | | I2T(Human) | 60.5 | 84.0 | 43.2 | 71.0 | 33.1 | 57.5 | 23.1 | 48.4 | 32.9 | 60.3 | 19.2 | 40.6 |
| + N:1 Aug | T2I(SD) | | 54.2 | 79.2 | 38.4 | 66.0 | 26.9 | 47.7 | 18.3 | 40.1 | 28.6 | 55.2 | 17.2 | 36.5 |
| (ours) MAT | | | 55.6 | 80.5 | 40.2 | 68.1 | 30.5 | 55.0 | 21.9 | 45.7 | 40.6 | 69.3 | 26.8 | 52.6 |
| + 1:N Aug | | I2T(OFA) | 54.5 | 80.1 | 39.3 | 67.3 | 30.3 | 54.2 | 21.7 | 45.6 | 39.0 | 67.8 | 26.0 | 51.2 |
| | | I2T(Human) | 57.7 | 81.5 | 41.4 | 69.2 | 32.3 | 56.3 | 22.8 | 47.3 | 42.6 | 70.7 | 28.0 | 54.1 |
| + N:1 Aug | T2T(SD) | | 51.9 | 77.9 | 36.3 | 64.1 | 26.3 | 48.5 | 18.5 | 40.4 | 36.9 | 65.4 | 24.3 | 48.9 |

## A.3 Ablation study for augmentations for MA$^2$T$_{Img}$.

Here, we present the ablation results on the effectiveness of augmentations. Figure 4, 5, 6, and 7 show the effectiveness of intra-modal and cross-modal augmentation techniques for MA$^2$T$_{Img}$. In Figures 5, 6, and 8, top indicates they are sorted decreasingly by alignment score. In Figure 7, the strength of augmentation levels ranges from 1 (weakest) to 5 (strongest), and is defined by three parameters {(minimum scale of random resizing, number of operations in RandAug, magnitude of RandAug)} as {(0.8, 0, 0), (0.8, 2, 3), (0.7, 2, 5), (0.6, 2, 7), (0.5, 2, 9)}, respectively.

---

[4]https://github.com/salesforce/ALBEF

[5]https://huggingface.co/stabilityai/stable-diffusion-2-1-base

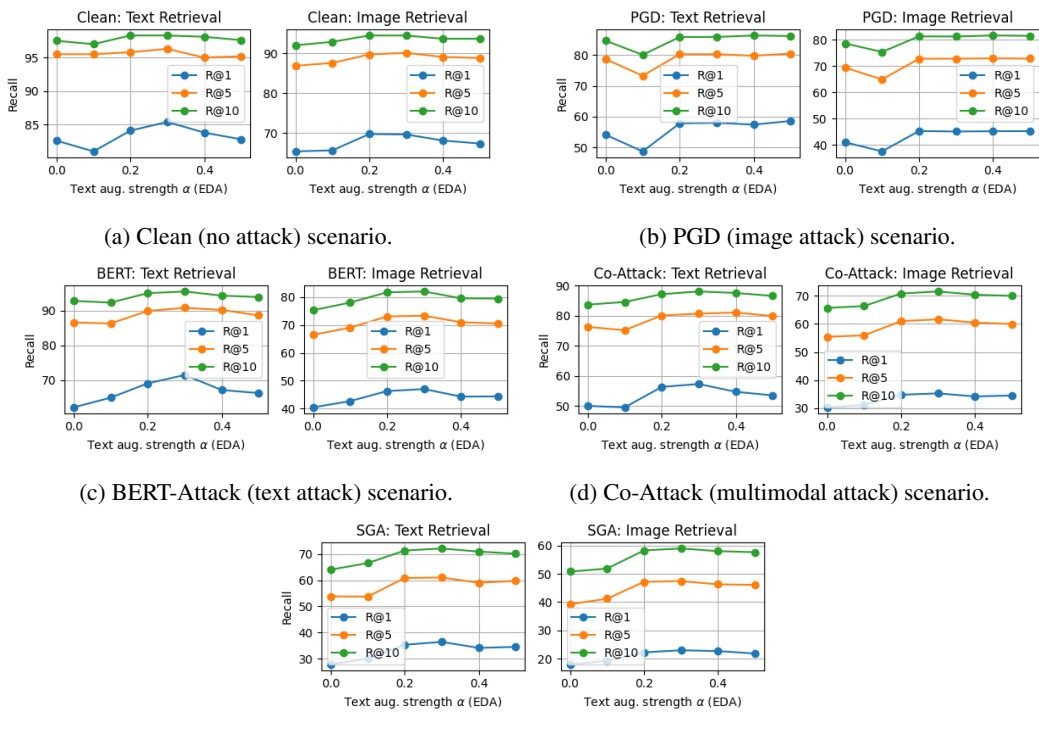

(a) Clean (no attack) scenario.

(b) PGD (image attack) scenario.

(c) BERT-Attack (text attack) scenario.

(d) Co-Attack (multimodal attack) scenario.

(e) SGA (multimodal attack) scenario

Figure 4: Effectiveness of intra-modal augmentation with EDA for enhancing adversarial robustness in VL models for ITR.

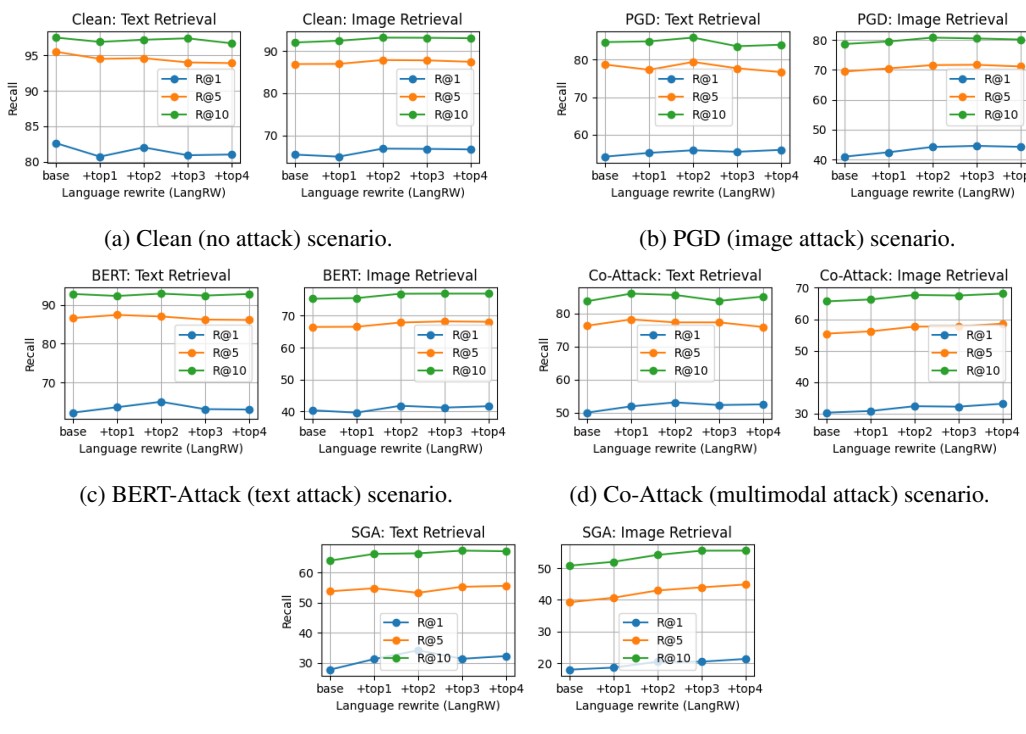

(a) Clean (no attack) scenario.

(b) PGD (image attack) scenario.

(c) BERT-Attack (text attack) scenario.

(d) Co-Attack (multimodal attack) scenario.

(e) SGA (multimodal attack) scenario.

Figure 5: Effectiveness of intra-modal augmentation with LangRW for $\text{MA}^2\text{T}_{\text{Img}}$.

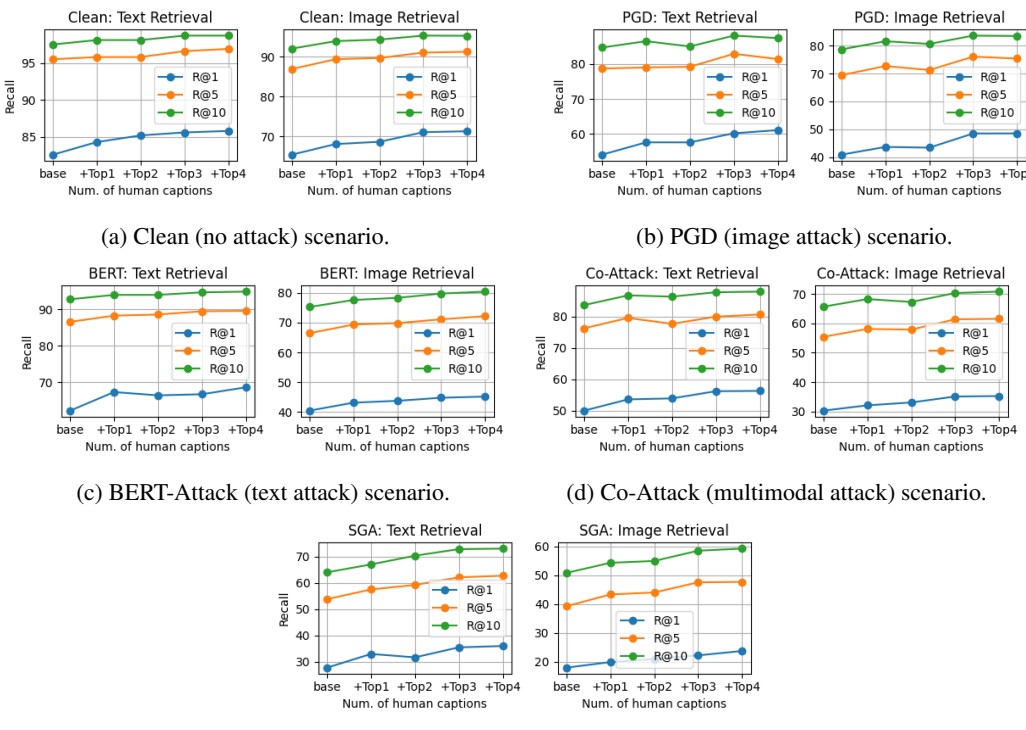

(a) Clean (no attack) scenario.

(b) PGD (image attack) scenario.

(c) BERT-Attack (text attack) scenario.

(d) Co-Attack (multimodal attack) scenario.

(e) SGA (multimodal attack) scenario.

Figure 6: Effectiveness of (ideal) cross-modal augmentation with ground truth captions for $\text{MA}^2\text{T}_{\text{Img}}$.

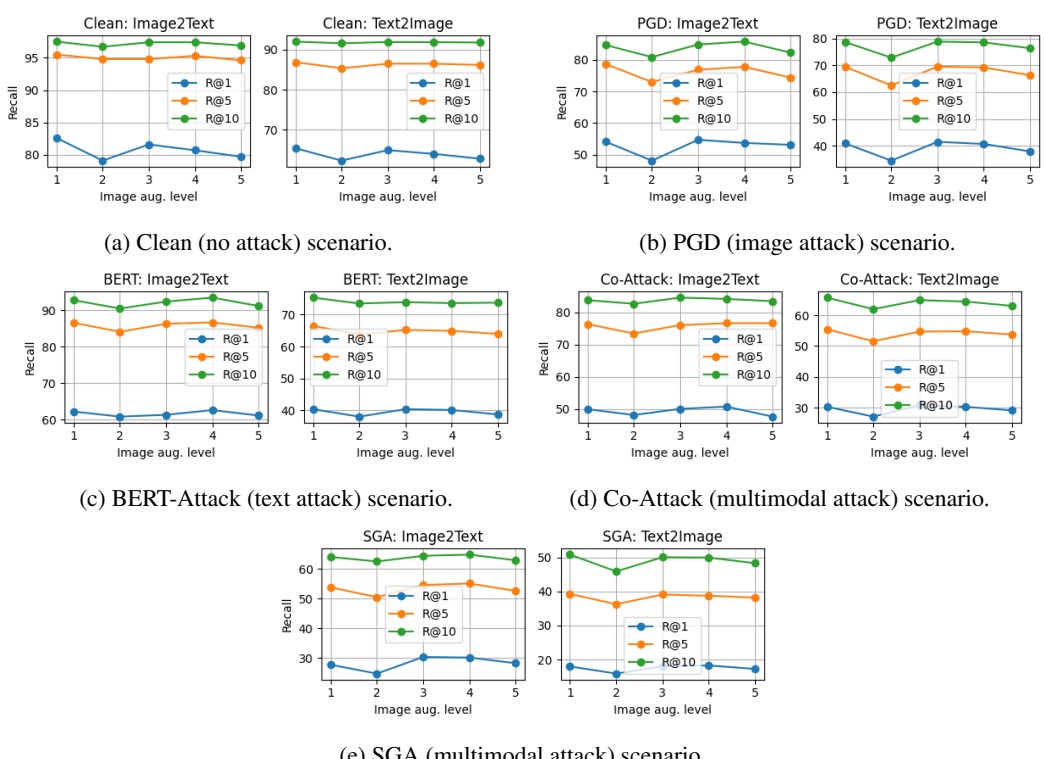

Figure 7: Effectiveness of intra-modal augmentation with RandAug for $\mathrm{MA^2T_{Img}}$.

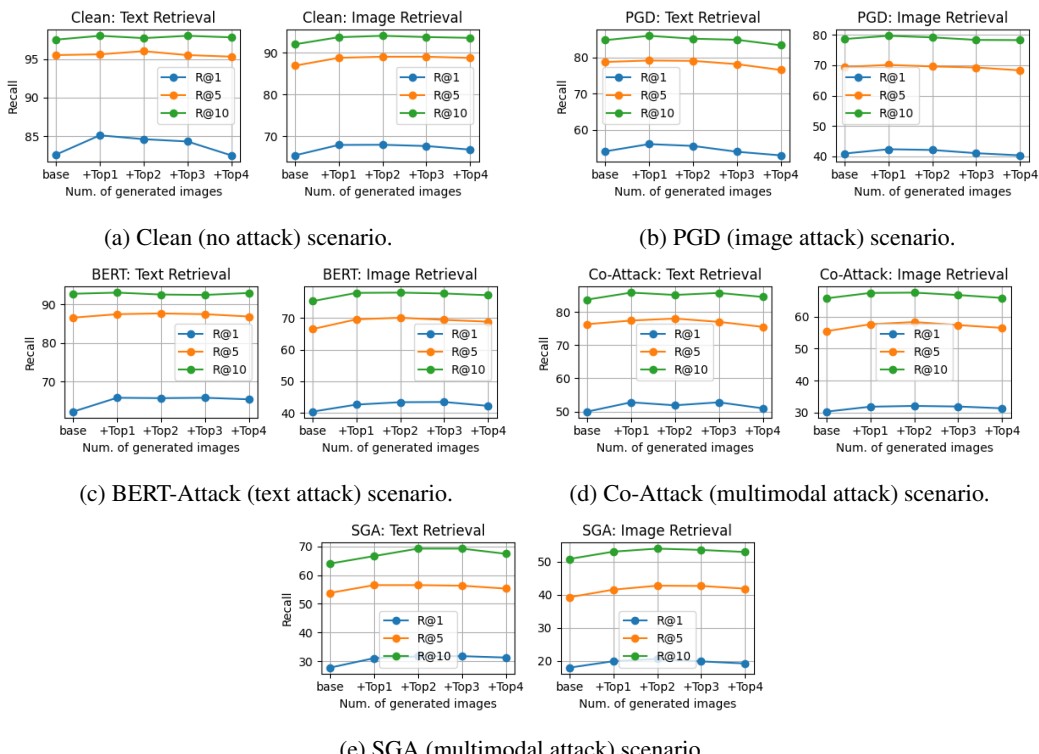

Figure 8: Effectiveness of cross-modal augmentation with SD for $\mathrm{MA^2T_{Img}}$.

