# OpenReview forum: "Leveraging One-To-Many Relationships in Multimodal Adversarial Defense for Robust Image-Text Retrieval"
_ICLR.cc/2025/Conference — Submitted to ICLR 2025_

### Official Review · Reviewer_5DW5 · 2024-11-01

**Soundness:** 2
**Presentation:** 2
**Contribution:** 2
**Rating:** 5
**Confidence:** 4

**Summary:**

This paper proposes a new defense framework, Multimodal Augmented Adversarial Training (MA2T), designed to enhance robustness in image-text retrieval tasks within vision-language models. MA2T is tailored for the CLIP model, leveraging one-to-many (1:N) image-text pairing and data augmentation to reduce the impact of multimodal adversarial attacks. This approach significantly improves model robustness on datasets such as Flickr30k and COCO.

**Strengths:**

1.	The authors are the first to propose a multimodal adversarial training method for ITR tasks, filling the research gap left by image-only defenses.
2.	Through an in-depth exploration of one-to-many relationships, the authors validate the effectiveness of various augmentation strategies, including text and image augmentation as well as cross-modal and unimodal augmentations.
3.	The experiments of the work show the operations make sense, and proposing data augmentation methods suitable for different tasks.
4.	The proposed framework can adapt to various real-world scenarios, providing a reference for AI security research.

**Weaknesses:**

1.	The experiments rely primarily on the Flickr30k and COCO datasets, lacking tests on other, more diverse real-world datasets.
2.	The framework is only tested on the CLIP model, without validation on other vision-language models, such as BLIP, to assess generalizability.
3.	There is a typo in the tenth line of the abstract; it seems the authors likely meant to write “comprehensive” rather than “conprehensive.”
4.	The paper lacks a clear framework diagram or visual results that would make the contributions of this work immediately understandable.

**Questions:**

1.	This paper lack a framework diagram, which limits its readability.
2.	In Table 3, the focus is mainly on comparing different augmentation strategies,  comparisons with other existing multimodal adversarial training methods are require.
3.	Why select Flickr30k and COCO datasets, it seems that the scenes in these two datasets are relatively limited?

---

> ### Author Response · Authors · 2024-11-24
> **Response by Authors**
>
> Thank you for taking the time to review our paper and your insights on it.
> We address the concerns raised below.
>
> ## [W1, Q3] Limited evaluation datasets.
>
> We humbly disagree. For the sake of coherence with the related work in adversarial robustness, we used the standard datasets Flickr30k and MSCOCO, as in Zhang et al.[A] and Lu et al.[B].
> To the best of our knowledge, these datasets are considered big and varied enough to validate adversarial attack and defense methods. If you have any specific recommendation for additional datasets that could enhance the evaluation or provide broader insights, we would greatly appreciate your suggestions.
>
> ## [W2] Other vision-language models
>
> We humbly disagree, as using CLIP alone is the standard in the related work on adversarial robustness. While we acknowledge that experiments on more models would enhance our claims, our primary focus in this work was to conduct **a thorough study on the effectiveness of diverse augmentation techniques in enhancing adversarial robustness in image retrieval tasks (ITR)**. To achieve this, we dedicated more computational resources to exploring data variations rather than model differences. This approach is consistent with the methodology of the related work TeCoA [C], which also focuses exclusively on CLIP-B/32 for detailed analysis. Following Zhang et al.[A] and Lu et al. [B], who studied adversarial attacks for ITR, we selected the CLIP-B/16 model, which is larger than CLIP-B/32. Although we are aware of the existence of other VL models such as ALBEF or BLIP, CLIP is still the most used backbone in vision-language research. Moreover, since their image-text matching is fundamentally similar, we believe that including more backbones would not lead to more additional findings other than a boost on the base performance.
>
> ## [W3] Typo
>
> Thank you so much for pointing this out. We have fixed it.
>
> ## [W4, Q1] Clear framework diagram and visual results
>
> Thank you for your constructive suggestion. We will include them in the camera ready.
>
>
> ## [Q2] Comparison with other multimodal adversarial training
>
> Since we are the pioneers in proposing multimodal adversarial training for ITR, there is no existing baseline method for direct comparison.
> Existing adversarial training methods for CLIP, such as TeCoA [C] (included in our experiments) and FARE [D], focus on defending against image-only attacks. Our method, MAT, is specifically designed for image-text multimodal attacks.
>
>
> ---
> References:
>
> [A] Zhang, Jiaming, Qi Yi, and Jitao Sang. "Towards adversarial attack on vision-language pre-training models." ACMMM 2022.
>
> [B] Lu, Dong, et al. "Set-level guidance attack: Boosting adversarial transferability of vision-language pre-training models." CVPR 2023.
>
> [C] Mao, Chengzhi, et al. "Understanding zero-shot adversarial robustness for large-scale models." ICLR 2023.
>
> [D] Schlarmann, Christian, et al. "Robust clip: Unsupervised adversarial fine-tuning of vision embeddings for robust large vision-language models." ICML 2024

---

### Official Review · Reviewer_ghV4 · 2024-11-01

**Soundness:** 3
**Presentation:** 2
**Contribution:** 3
**Rating:** 5
**Confidence:** 3

**Summary:**

This paper introduces Multimodal Augmented Adversarial Training (MA2T) to improve adversarial robustness in image-text retrieval (ITR). Extending beyond unimodal defenses, MA2T combines image and text perturbations and incorporates one-to-many and many-to-one augmentations to counteract overfitting and enhance multimodal resilience. Experiments on Flickr30k and COCO validate that MA2T improves robustness, especially with cross-modal augmentations.

**Strengths:**

The paper addresses adversarial robustness in image-text retrieval (ITR) by employing multimodal adversarial training alongside one-to-many and many-to-one augmentations. This approach leverages the multimodal characteristics of ITR data to enhance defenses against attacks. The experimental methodology is robust, featuring well-structured evaluations on Flickr30k and COCO, which illustrate the advantages of cross-modal augmentations. Overall, the work is clearly articulated, providing sufficient context and explanations, and is relevant for advancing robust vision-language models in the expanding field of multimodal research.

**Weaknesses:**

A substantive assessment of the weaknesses of the paper. Focus on constructive and actionable insights on how the work could improve towards its stated goals. Be specific, avoid generic remarks. For example, if you believe the contribution lacks novelty, provide references and an explanation as evidence; if you believe experiments are insufficient, explain why and exactly what is missing, etc.

While the paper proposes a promising approach, several areas need improvement to strengthen claims of broader applicability and robustness. First, the experiments are limited to CLIP as the only vision-language model, which restricts conclusions about model generalizability. Evaluating the framework on additional models, such as BLIP or ALBEF, would provide a more thorough understanding of its robustness across various architectures. Additionally, the current augmentation strategy for image perturbations may introduce distribution shifts that could negatively affect performance. Finally, although the paper discusses the limitations of unimodal defenses in a multimodal context, a more comprehensive theoretical analysis of why cross-modal augmentations specifically enhance ITR robustness is warranted.

**Questions:**

Since the framework is tested solely on CLIP, do the authors foresee challenges in adapting MA2T to other vision-language models, such as BLIP or ALBEF?

The paper notes that image augmentations may introduce distribution shifts that could affect performance. Have the authors investigated alternative augmentation techniques or constraints to mitigate this impact?

Minor issue:
Some Grammatical mistakes are there – like “conprehensive” instead of “comprehensive” [line 021]. “mutlimodal” instead of “multimodal” [line 225]. A thorough proofreading will be helpful.

---

> ### Author Response · Authors · 2024-11-24
> **Response by Authors (1/2)**
>
> Thank you for taking the time to review our paper and your insights on it.
> We address the concerns raised below.
>
> ## [W1] Using only CLIP is a limitation
>
> We humbly disagree, as using CLIP alone is the standard in the related work on adversarial robustness. While we acknowledge that experiments on more models would enhance our claims, our primary focus in this work was to conduct **a thorough study on the effectiveness of diverse augmentation techniques in enhancing adversarial robustness in image retrieval tasks (ITR)**. To achieve this, we dedicated more computational resources to exploring data variations rather than model differences. This approach is consistent with the methodology of the related work TeCoA [C], which also focuses exclusively on CLIP-B/32 for detailed analysis. Following Zhang et al.[D] and Lu et al. [E], who studied adversarial attacks for ITR, we selected the CLIP-B/16 model, which is larger than CLIP-B/32. Although we are aware of the existence of other VL models such as ALBEF or BLIP, CLIP is still the most used backbone in vision-language research. Moreover, since their image-text matching is fundamentally similar, we believe that including more backbones would not lead to more additional findings other than a boost on the base performance.
>
> ## [W2] Limitations in image augmentation
>
> We do not consider this a weakness in our study, but rather, a novel finding.
> The fact that simple text augmentations provide higher adversarial robustness than image augmentations is particularly noteworthy and, to the best of our knowledge, has not been proved before. One of our contributions is studying the effectiveness of different augmentations and, for the first time, providing empirical results that show how the diversity and alignment of the augmentations are two key axis to achieve adversarial robustness.
>
> The cause for lower performance in image augmentations is analyzed in the paper (Line 462-) as follows:
> > This is because generating image augmentations that do not lack diversity but also do not deviate significantly from the original data distribution is more challenging due to the high dimensionality of the image space. On the other hand, text modality is more amenable to augmentation, as the text space is lower-dimensional and more structured, making it easier to generate appropriate diversity in the augmented data points.
>
>
> ## [W3] A more comprehensive theoretical analysis of why cross-modal augmentations specifically enhance ITR robustness.
>
> We regret not providing a theoretical analysis in our work. However, we believe that our claims are adequately supported by the thorough empirical results presented.
>
> Theoretically, data augmentation in adversarial training mitigates robust overfitting by enhancing generalization to unseen data [A], for example adversarially perturbed data. Our work follows this very same theory, when applied to more than one modality.
> Empirically, this has been validated in one-to-one unimodal tasks (i.e., image classification). For example, Wang et al. [B] demonstrated that synthetic images generated by diffusion models can improve adversarial robustness. However, as shown in our experiments, the unimodal image augmentations of the related work underperform in the multimodal task of image-text retrieval, where perturbations can occur in both modalities.
> While [B] highlighted the role of "better" diffusion models (in terms of image quality) in enhancing robustness, our work further explores this analysis to vision-language robustness. Specifically, we provide novel insights into what constitutes "better" augmentations in vision-language multi-modal adversarial training: augmentations that maintain high image-text alignment and ensure sufficient diversity of image-text pairs.
>
> If you could provide more detailed suggestions on the theoretical analysis or specific aspects that we should prove, we would be happy to consider them.
>
> ## [Q1] Would there be a challenges in adapting MA2T to BLIP or ALBEF?
>
> Since BLIP and ALBEF share fundamentally similar image-text matching mechanisms with contrastive loss, akin to CLIP, there is no reason to think that MA2T would not be effective with these models.

---

> > ### Author Response · Authors · 2024-11-24
> > **Response by Authors (2/2)**
> >
> > ## [Q2] Alternative image augmentation techniques
> >
> > In addition to the experiments provided in the paper, we evaluated a version of Stable Diffusion (SD) that was fine-tuned using LoRA on the Flickr30k dataset. While we observed a slight improvement in performance, it did not surpass the performance of text augmentations.
> > Our results indicate that, obtaining further performance gains requires better alignment with the original samples, and thus, a more sophisticated fine-tuning approach for SD seems necessary. However, this time we considered it was out of the scope of our paper.
> >
> > ## [Q3] Minor typos
> >
> > Thank you so much for pointing these out. We have fixed them.
> >
> > ---
> > References:
> >
> > [A] Rebuffi, S. A., Gowal, S., Calian, D. A., Stimberg, F., Wiles, O., & Mann, T. A. "Data augmentation can improve robustness." NeurIPS 2021.
> >
> > [B] Wang, Zekai, et al. "Better diffusion models further improve adversarial training." ICML 2023.
> >
> > [C] Mao, Chengzhi, et al. "Understanding zero-shot adversarial robustness for large-scale models." ICLR 2023.
> >
> > [D] Zhang, Jiaming, Qi Yi, and Jitao Sang. "Towards adversarial attack on vision-language pre-training models." ACMMM 2022.
> >
> > [E] Lu, Dong, et al. "Set-level guidance attack: Boosting adversarial transferability of vision-language pre-training models." CVPR 2023.

---

### Official Review · Reviewer_oUBz · 2024-11-01

**Soundness:** 3
**Presentation:** 3
**Contribution:** 2
**Rating:** 3
**Confidence:** 3

**Summary:**

This paper explored adversarial attack and defense for image-text retrieval (ITR) using vision-language models. It proposed Multimodal Augmented Adversarial Training (MA2T), using one-to-many relationships in image-text pairs to improve model robustness. The authors claimed improvements in adversarial robustness, especially when using text augmentations over image perturbations.

**Strengths:**

- An interesting problem of multimodal adversarial defense, particularly for ITR.
- The paper proposed a new defense strategy, MA2T, to improve robustness by incorporating multimodal adversarial training and augmentation.
- The paper conducted many experiments across multiple attack types, with detailed augmentation analysis.

**Weaknesses:**

- It seems unclear why one-to-many augmentations should directly improve adversarial robustness in ITR, it would be good to add some theoretical explanations if possible.
- Following the above, the selection choice, including the multimodal training setup, appears empirically driven without a theoretical basis.
- The paper used CLIP-ViT-B/16 as the base model and reported improvements in robustness metrics (e.g., 1.7%–8.7%). The authors should have realized that CLIP-ViT-B/16 is quite a small model, and the performance improvement on this may not be generalized to a larger model, which is said, the large model may already show much better adversarial robustness than the small model. So it is recommended to conduct a study on larger models to see the performance and the improvement gain compared with small models,
- The paper only used a base model. Though many attacks have been studied, it seems unclear whether the proposed method only works on the models with architectures like CLIP or can be generalized to other model architectures. It is recommended that other model architectures be investigated as well.
- Evaluations are limited to Flickr30k and COCO datasets. Existing studies have shown that Flickr is quite a simple dataset, so it is recommended that other, more complex datasets be explored.
- Some evaluations show selective use of augmented pairs, while others apply them inconsistently across attack types and scenarios. This inconsistency may lead to ambiguity around the robustness gains attributable to MA2T.

**Questions:**

Please see the comments above.

---

> ### Author Response · Authors · 2024-11-24
> **Response by Authors (1/2)**
>
> Thank you for taking the time to review our paper and your insights on it.
> We address the concerns raised below.
>
> ## [W1] Theory on why one-to-many augmentations improve adversarial robustness
>
> Theoretically, data augmentation in adversarial training mitigates robust overfitting by enhancing generalization to unseen data [F], for example adversarially perturbed data. Our work follows this very same theory, when applied to more than one modality.
> Empirically, this has been validated in one-to-one unimodal tasks (i.e., image classification). For example, Wang et al. [A] demonstrated that synthetic images generated by diffusion models can improve adversarial robustness. However, as shown in our experiments, the unimodal image augmentations of the related work underperform in the multimodal task of image-text retrieval, where perturbations can occur in both modalities.
> While [A] highlighted the role of "better" diffusion models (in terms of image quality) in enhancing robustness, our work further explores this analysis to vision-language robustness. Specifically, we provide novel insights into what constitutes "better" augmentations in vision-language multi-modal adversarial training: augmentations that maintain high image-text alignment and ensure sufficient diversity of image-text pairs.
>
> ## [W2] Multimodal training setup appears empirically driven without a theoretical basis.
>
> We apologize for the lack of clarity in our explanation.
> Our multimodal training framework simply builds upon the standard adversarial training approach based on the min-max optimization principle [B], but extends to vision-language models. Here, the adversarial attack in image classification [B] maximizes the loss function, while the model minimizes it:
>
> $min_{\theta}  \ \rho(\theta), \ \text{where} \ \rho(\theta) = E_{(x,y) \sim \mathcal{D}} \left[ \max_{\delta \in \mathcal{S}} L(\theta, x + \delta, y) \right].$
>
> Extending this formulation to vision-language models, we define the general objective as follows:
>
> $min_{\theta} \ \rho(\theta), \ \text{where} \ \rho(\theta) = {E}_{(x,y) \sim \mathcal{D}} \left[ \max  L(\theta, x + \delta_x, t+ \delta_y) \right].$
>
> Our **Multimodal Adversarial Training (MAT)** framework addresses the inner maximization problem using the approximation detailed in Section 3.2. Specifically, our method adopts a simple yet effective strategy: first updating the text modality and then the image modality. While this sequential approach provides an effective baseline, further improvements could be explored by iteratively updating image and text modalities to better maximize the loss function. We leave this refinement for future work.
>
> ## [W3,W4] Regarding the use of larger models and other model architecture.
>
> We humbly disagree, as using CLIP alone is the standard in the related work on adversarial robustness. While we acknowledge that experiments on more models would enhance our claims, our primary focus in this work was to conduct **a thorough study on the effectiveness of diverse augmentation techniques in enhancing adversarial robustness in image retrieval tasks (ITR)**. To achieve this, we dedicated more computational resources to exploring data variations rather than model differences. This approach is consistent with the methodology of the related work TeCoA [C], which also focuses exclusively on CLIP-B/32 for detailed analysis. Following Zhang et al.[D] and Lu et al. [E], who studied adversarial attacks for ITR, we selected the CLIP-B/16 model, which is larger than CLIP-B/32. Although we are aware of the existence of other VL models such as ALBEF or BLIP, CLIP is still the most used backbone in vision-language research. Moreover, since their image-text matching is fundamentally similar, we believe that including more backbones would not lead to more additional findings other than a boost on the base performance.

---

> > ### Author Response · Authors · 2024-11-24
> > **Response by Authors (2/2)**
> >
> > ## [W5] Limited evaluation datasets.
> >
> > For the sake of coherence with the related work in adversarial robustness, we used the standard datasets Flickr30k and MSCOCO, as in Zhang et al.[D] and Lu et al.[E].
> > To the best of our knowledge, these datasets are considered big and varied enough to validate adversarial attack and defense methods. If you have any specific recommendation for additional datasets that could enhance the evaluation or provide broader insights, we would greatly appreciate your suggestions.
> >
> > ## [W6] Some evaluations show selective use of augmented pairs, while others apply them inconsistently across attack types and scenarios. This inconsistency may lead to ambiguity around the robustness gains attributable to MA2T.
> >
> > Could you kindly clarify this question for us? We are sure that our setting is coherent among experiments and ablations. We apologize for any confusion and appreciate your understanding.
> >
> > ---
> > References:
> >
> > [A] Wang, Zekai, et al. "Better diffusion models further improve adversarial training." ICML 2023.
> >
> > [B] Madry, Aleksander. "Towards deep learning models resistant to adversarial attacks." ICLR 2018.
> >
> > [C] Mao, Chengzhi, et al. "Understanding zero-shot adversarial robustness for large-scale models." ICLR 2023.
> >
> > [D] Zhang, Jiaming, Qi Yi, and Jitao Sang. "Towards adversarial attack on vision-language pre-training models." ACMMM 2022.
> >
> > [E] Lu, Dong, et al. "Set-level guidance attack: Boosting adversarial transferability of vision-language pre-training models." CVPR 2023.
> >
> > [F] Rebuffi, S. A., Gowal, S., Calian, D. A., Stimberg, F., Wiles, O., & Mann, T. A. "Data augmentation can improve robustness." NeurIPS 2021.

---

> > > ### Comment · Reviewer_oUBz · 2024-12-02
> > > **Thank you for the response**
> > >
> > > The reviewer appreciates the authors’ efforts in providing a response. However, the current version of the paper lacks sufficient experiments on additional models for benchmarking and the use of a larger model to demonstrate effectiveness, which weakens the support for the claims made. Furthermore, the requested experimental results were not provided in the response, possibly due to time constraints preventing the completion of these experiments. The reviewer suggests that the authors revise the paper to strengthen its theoretical foundation and include more comprehensive experiments to enhance its quality.

---

> > > > ### Author Response · Authors · 2024-12-02
> > > > **Response by Authors**
> > > >
> > > > We sincerely appreciate your response.
> > > >
> > > > We would like to refer the reviewer to our comments above regarding why including more models is not essential to our work. We emphasize that the standard settings for adversarial defense papers have been to focus only on the CLIP model and conduct adversarial training on a single dataset, as seen in TeCoA[ICLR'23], FARE[ICML'24], PMG-AFT[CVPR'24], and TGA-ZSR[NeurIPS'24]. This is due to the computational complexity of adversarial training strategies. In contrast to adversarial attack papers, which are less computationally demanding, defense papers require significantly more resources. Our study, however, includes over 20 adversarially trained models to provide a comprehensive analysis of augmentation techniques, offering substantial evidence for our claims. While we agree that trying more architectures could be a nice addition, there is no intuition on them resulting in critical insights regarding our work. Thus, we would appreciate clarification on why our work must extend to more models and the name of those models, different from those considered in the four referenced papers above.
> > > >
> > > > Similarly, we provided references regarding the theoretical foundation of our idea, so we kindly ask you to provide any insight on why that theory is invalid if you believe so. Otherwise, we cannot provide a rebuttal that helps the area chair to understand the value of our paper.
> > > >
> > > > Since your response refers only to those two points, we assume that your other concerns (W5, W6) were solved. In that case, we kindly ask you to revise your score.
> > > >
> > > > ---
> > > > **Reference**
> > > > [TeCoA] Mao, Chengzhi, et al. "Understanding zero-shot adversarial robustness for large-scale models." ICLR'23
> > > > [FARE] Schlarmann, Christian, et al. "Robust clip: Unsupervised adversarial fine-tuning of vision embeddings for robust large vision-language models." ICML'24
> > > > [PMG-AFT] Wang, Sibo, et al. "Pre-trained model guided fine-tuning for zero-shot adversarial robustness." CVPR'24
> > > > [TGA-ZSR] Yu, Lu, Haiyang Zhang, and Changsheng Xu. "Text-Guided Attention is All You Need for Zero-Shot Robustness in Vision-Language Models." NeurIPS'24

---

### Official Review · Reviewer_NoRj · 2024-11-18

**Soundness:** 3
**Presentation:** 2
**Contribution:** 2
**Rating:** 5
**Confidence:** 3

**Summary:**

This research introduces novel defense strategies for Image-Text Retrieval (ITR) by addressing the limitations of existing methods tailored for image classification. A pioneering approach is demonstrated, emphasizing the significance of multimodal adversarial training in enhancing the robustness of ITR systems against diverse attacks. Furthermore, a comprehensive analysis of leveraging one-to-many relationships is conducted, revealing the efficacy of diverse augmentations across image and text modalities for bolstering the resilience of ITR models.

**Strengths:**

1.This research pioneers a new direction in defense strategies for ITR, highlighting the inadequacies of conventional image classification defense methods.
2.The introduction of multimodal adversarial training significantly improves the robustness of ITR systems.
3.This study offers an in-depth analysis of leveraging one-to-many relationships
4.Well-written and easy to read.

**Weaknesses:**

1. Both the selection of datasets and the methodological exposition in this work are relatively weak and lack persuasiveness. It is suggested that the authors should not confine themselves to COCO and Flickr datasets but also test on more diverse datasets, such as remote sensing scenes, to thoroughly validate the generalizability of the proposed method. Furthermore, the introduced method lacks sufficient theoretical justification.
2. The ablation experiments are too simplistic; at the very least, different visual-language foundation models should be subjected to ablation analysis.

**Questions:**

1. Can the proposed method be extended to tasks at a finer granularity, such as VL segmentation and detection?
2. Since text augmentation is superior to image augmentation, is there a similar conclusion for the audio modality as well?
3. Since the author mentioned a one-to-many strategy, what about many-to-many strategies, such as [1]?

[1]. Leveraging Many-To-Many Relationships for Defending Against Visual-Language Adversarial Attacks, arXiv 2024

---

> ### Author Response · Authors · 2024-11-24
> **Response by Authors (1/2)**
>
> Thank you for taking the time to review our paper and your insights on it.
> We address the concerns raised below.
>
> ## [W1-1] Limited evaluation datasets.
>
> Flickr30k and MSCOCO are the standard image-text retrieval (ITR) datasets used for evaluation of adversarial attacks. This the case of the related work of Zhang et al.[A] and Lu et al.[B], which we follow for consistency. These datasets contain a variety of scenes that is wide enough to evaluate our method.
>
> While remote sensing datasets are indeed used in some ITR scenarios, we do not see them essential for proving the validity of our method, as they are tied to very specific applications such as environmental monitoring and agriculture. Furthermore, the image/text augmentations that can be applied to remote sensing data are very limited, and standard text2image and image2text methods are not tuned to that specific domain.
>
> To summarize, remote sensing datasets are not commonly used in adversarial attack research, and we do not consider them essential to evaluate our method. If you have any recommendations that could enhance the evaluation or provide a broader perspective in our adversarial defense scenario, we would greatly appreciate your suggestions and the reason why.
>
> ## [W1-2] Introduced method lacks sufficient theoretical justification.
>
> ### 1. Justification of multimodal adversarial training
> Our multimodal training framework simply builds upon the standard adversarial training approach based on the min-max optimization principle [C], but extends to vision-language models. Here, the adversarial attack in image classification [C] maximizes the loss function, while the model minimizes it:
>
> $min_{\theta}  \ \rho(\theta), \ \text{where} \ \rho(\theta) = E_{(x,y) \sim \mathcal{D}} \left[ \max_{\delta \in \mathcal{S}} L(\theta, x + \delta, y) \right].$
>
> Extending this formulation to vision-language models, we define the general objective as follows:
>
> $min_{\theta} \ \rho(\theta), \ \text{where} \ \rho(\theta) = {E}_{(x,y) \sim \mathcal{D}} \left[ \max  L(\theta, x + \delta_x, t+ \delta_y) \right].$
>
> Our **Multimodal Adversarial Training (MAT)** framework addresses the inner maximization problem using the approximation detailed in Section 3.2. Specifically, our method adopts a simple yet effective strategy: first updating the text modality and then the image modality. While this sequential approach provides an effective baseline, further improvements could be explored by iteratively updating image and text modalities to better maximize the loss function. We leave this refinement for future work.
>
> ### 2. Justification of augmentation for robustness
>
> Theoretically, data augmentation in adversarial training mitigates robust overfitting by enhancing generalization to unseen data [D], for example adversarially perturbed data. Our work follows this very same theory, when applied to more than one modality.
> Empirically, this has been validated in one-to-one unimodal tasks (i.e., image classification). For example, Wang et al. [E] demonstrated that synthetic images generated by diffusion models can improve adversarial robustness. However, as shown in our experiments, the unimodal image augmentations of the related work underperform in the multimodal task of image-text retrieval, where perturbations can occur in both modalities.
> While [E] highlighted the role of "better" diffusion models (in terms of image quality) in enhancing robustness, our work further explores this analysis to vision-language robustness. Specifically, we provide novel insights into what constitutes "better" augmentations in vision-language multi-modal adversarial training: augmentations that maintain high image-text alignment and ensure sufficient diversity of image-text pairs.

---

> > ### Author Response · Authors · 2024-11-24
> > **Response by Authors (2/2)**
> >
> > ## [W2] The ablation experiments are too simplistic; at the very least, different visual-language foundation models should be subjected to ablation analysis.
> >
> > We humbly disagree, as using CLIP alone is the standard in the related work on adversarial robustness. While we acknowledge that experiments on more models would enhance our claims, our primary focus in this work was to conduct **a thorough study on the effectiveness of diverse augmentation techniques in enhancing adversarial robustness in image retrieval tasks (ITR)**. To achieve this, we dedicated more computational resources to exploring data variations rather than model differences. This approach is consistent with the methodology of the related work TeCoA [F], which also focuses exclusively on CLIP-B/32 for detailed analysis. Following Zhang et al.[A] and Lu et al. [B], who studied adversarial attacks for ITR, we selected the CLIP-B/16 model, which is larger than CLIP-B/32. Although we are aware of the existence of other VL models such as ALBEF or BLIP, CLIP is still the most used backbone in vision-language research. Moreover, since their image-text matching is fundamentally similar, we believe that including more backbones would not lead to more additional findings other than a boost on the base performance.
> >
> > ## [Q1] Can the proposed method be extended to tasks at a finer granularity, such as VL segmentation and detection?
> >
> > Thank you for pointing this out. VL segmentation and detection also leverage image-text matching models, such as CLIP, as a backbone to enable understanding of the image-text relationship. Thus, we believe our method is also applicable to enhance robustness in these tasks. Although these tasks fall out of the scope of adversarial robustness in image-text retrieval, they are an important direction for future work that continues this line of research.
> >
> > ## [Q2] Since text augmentation is superior to image augmentation, is there a similar conclusion for the audio modality as well?
> >
> > While audio modality is out-of-scope of our research in image-text retrieval, this is a very interesting direction to analyze. Thank you so much for your valuable suggestion.
> > We presume that, since the audio modality is also high-dimensional and sequential, it is likely to encounter similar challenges, such as audio generation models producing data with a distribution that deviates too much from the training data.
> >
> > ## [Q3] Since the author mentioned a one-to-many strategy, what about many-to-many strategies, such as [1]?
> >
> > Since current image augmentations suffer from having a distribution too different from the training data, a many-to-many strategy combining image and text augmentation may not lead to performance improvements, as indicated in [1]. It is also important to note that combining image and text augmentations requires a careful balance between alignment and diversity of each augmentation, which complicates the selection of appropriate augmentations. We leave this analysis to future work.
> >
> > ---
> > References:
> >
> > [A] Zhang, Jiaming, Qi Yi, and Jitao Sang. "Towards adversarial attack on vision-language pre-training models." ACMMM 2022.
> >
> > [B] Lu, Dong, et al. "Set-level guidance attack: Boosting adversarial transferability of vision-language pre-training models." CVPR 2023.
> >
> > [C] Madry, Aleksander. "Towards deep learning models resistant to adversarial attacks." ICLR 2018.
> >
> > [D] Rebuffi, S. A., Gowal, S., Calian, D. A., Stimberg, F., Wiles, O., & Mann, T. A. "Data augmentation can improve robustness." NeurIPS 2021.
> >
> > [E] Wang, Zekai, et al. "Better diffusion models further improve adversarial training." ICML 2023.
> >
> > [F] Mao, Chengzhi, et al. "Understanding zero-shot adversarial robustness for large-scale models." ICLR 2023.

---

### Author Response · Authors · 2024-11-24
**General Response by Authors**

We appreciate all the reviewers for taking the time to review our paper and for their thoughtful efforts.

Thank you for considering our work a pioneer [NoRj, 5DW5], interesting [oUBz], well-written [NoRj, ghV4], with an experimental setting that is robust [ghV4, 5DW5], well-structured [ghV4] and in-depth [NoRj, oUBz, 5DW5]. We appreciate the feedback and will clarify all points in the camera-ready version.

First, we would like to clarify our contributions:
- We propose **a new defense paradigm** for adversarial robustness in the multimodal task of image-text retrieval.
- **For the first time, we explore the benefits of standard image and text augmentation techniques** for adversarial robustness in various combinations.
- We analyze the effectiveness of different augmentations and, for the first time, **provide empirical evidence revealing the importance of diversity and alignment of augmentations for robustness**.

However, the final scores do not reflect these strengths and mostly focus on requiring additional experiments that, in our humble opinion, do not seem essential. **Our method has been validated following the standard settings of the related work regarding datasets and models**, and we humbly request the reviewers reconsider their scores.

Key points of disagreement (discussed more in-depth for each author):

- *Additional datasets:* **Flickr and COCO are the standard datasets** evaluating adversarial robustness in the related works, providing diverse general scenes.
- *Additional models:* **Our primary focus was to thoroughly study** the impact of diverse augmentation techniques on adversarial robustness, and dedicated computational resources to exploring data variations. The defense method, TeCoA [A], also focuses exclusively on CLIP-B/32 for detailed analysis since adversarial training is computationally expensive. Moreover, since the other models mentioned (i.e., ALBEF, BLIP) are based on the same multimodal contrastive learning as CLIP,  evaluating ALBEF and BLIP may not provide any critical insight regarding the effectiveness of multimodal augmentations, apart from a boost in CLIP’s base performance. Instead, we focused on thoroughly evaluating various attack methods and augmentations.
- *Lack of theoretical background:* Our work is **based on a well-established theory for data augmentation in adversarial robustness [B]**. We demonstrate that unimodal methods behind this theory underperform in cross-modal retrieval, and demonstrated that their alignment and diversity are the key factors.

While we also believe that adding more experiments would be nice, in our humble opinion, none of them are essential points that debunk our contributions in a way that the paper deserves to be rejected. We focused our efforts on evaluating a variety of augmentations and attack methods. **Increasing the number of models and datasets in this study would exponentially increase the number of required combinations unreasonably, given that training adversarial defense methods are computationally expensive.**

**We also ask reviewers to specify the name of the suggested datasets or models and the specific reason why they are essential**, as it is hard to provide an answer otherwise.

We apologize for any unclear points and will address them in the camera-ready version. We kindly ask for a reassessment of our work.

---
[A] Mao, Chengzhi, et al. "Understanding zero-shot adversarial robustness for large-scale models." ICLR 2023.

[B] Rebuffi, S. A., Gowal, S., Calian, D. A., Stimberg, F., Wiles, O., & Mann, T. A. "Data augmentation can improve robustness." NeurIPS 2021.

---

> ### Author Response · Authors · 2024-11-28
> **A Kind Reminder by Authors**
>
> Dear Reviewers,
>
> Thank you once again for taking the time to review our paper. We greatly value your insightful comments and sincerely appreciate your efforts.
>
> We kindly request that you review our replies. Your feedback is invaluable in addressing concerns and improving our work.
>
> If anything remains unclear, please don’t hesitate to reach out. We would be glad to provide further clarification.
>
> Thank you sincerely, Authors

---

### Meta-Review · Area_Chair_RcrD · 2024-12-20

**Metareview:**

This paper investigates adversarial attack and defense for image-text retrieval (ITR) tasks. To tackle this problem, this paper introduces Multimodal Augmented Adversarial Training (MA2T) awaring of one-to-many image-text correspondences and using diverse augmentation techniques. Experimental results show that the proposed method can enhance adversarial robustness of ITR models, especially when using cross-modal augmentation.

This paper has four negative initial reviews, while only one reviewer responded to the authors despite the reminder by the authors and the AC. I made my decision not solely based on the initial scores, but based on the diverse perspectives, including the authors' response.
The reviewers have three shared concerns:

- [NoRj, oUBz, 5DW5] The evaluation dataset is only limited to COCO and Flickr30k which are known to be simple datasets.
- [NoRj, oUBz, ghV4] There is no theoretical justification of why one-to-many augmentation can help adversarial robustness (it is different from the theoretical justification of minimax optimization, which is widely known theory in adversarial robustness of classification tasks). There could be a potential risk of this augmentation strategy, but there is no related discussion or theory.
- [NoRj, oUBz, ghV4, 5DW5] The study is only based on a single model, CLIP ViT-B/16. We may need more diverse backbones, such as BLIP, ALBEF, or larger backbones, such as CLIP ViT-L/14.

I agree with the reviewers' point. Although the authors mentioned that they followed the previous work, testing more backbones will be important for this submission. For example, [A] showed that different VLMs behave in different ways even under the same evaluation scenario (some models are good at recall, some models are good at precision). Similar findings could be observed in this scenario. I personally recommend adding more backbones, such as [1] different CLIP backbones [2] BLIP backbone [3] image-text cross-modal retrieval models based on triplet loss, e.g., VSE infinity [B]

- [A] Chun, Sanghyuk, et al. "Eccv caption: Correcting false negatives by collecting machine-and-human-verified image-caption associations for ms-coco." European Conference on Computer Vision. Cham: Springer Nature Switzerland, 2022.
- [B] Chen, Jiacheng, et al. "Learning the best pooling strategy for visual semantic embedding." Proceedings of the IEEE/CVF conference on computer vision and pattern recognition. 2021.

I partially agree with the authors on evaluation datasets. There are not many evaluation datasets for image-text cross-modal retrieval. One possible alternative is ECCV Caption [A] or CxC [C], which are the extended versions of COCO Caption with many-to-many correspondences.

- [C] Parekh, Zarana, et al. "Crisscrossed captions: Extended intramodal and intermodal semantic similarity judgments for MS-COCO." arXiv preprint arXiv:2004.15020 (2020).

Finally, the most significant issue of this paper is the lack of understanding of why and how the proposed one-to-many augmentation enhances adversarial robustness. Although the authors mentioned general theories for adversarial robustness and adversarial training, they cannot directly tackle the one-to-many augmentation.

Overall, I think this paper needs more empirical analyses (more backbones, more benchmarks) and more theoretical or high-level insights (how the proposed augmentation works for adversarial robustness).

**Additional Comments On Reviewer Discussion:**

The reviewers have three shared concerns:

- [NoRj, oUBz, 5DW5] The evaluation dataset is only limited to COCO and Flickr30k which are known to be simple datasets.
- [NoRj, oUBz, ghV4] There is no theoretical justification of why one-to-many augmentation can help adversarial robustness (it is different from the theoretical justification of minimax optimization, which is widely known theory in adversarial robustness of classification tasks). There could be a potential risk of this augmentation strategy, but there is no related discussion or theory.
- [NoRj, oUBz, ghV4, 5DW5] The study is only based on a single model, CLIP ViT-B/16. We may need more diverse backbones, such as BLIP, ALBEF, or larger backbones, such as CLIP ViT-L/14.

Reviewer oUBz disagrees with the authors' response, and keeps their initial negative rating.

---

### Decision · Program_Chairs · 2025-01-22

Reject